# In-phase millennial-scale glacier changes in the tropics and North Atlantic regions during the Holocene

V. Jomelli [1,2 ✉], D. Swingedouw [3], M. Vuille[4], V. Favier [5], B. Goehring[6], J. Shakun [7], R. Braucher [2], I. Schimmelpfennig[2], L. Menviel [8], A. Rabatel [5], L. C. P. Martin [9,10], P.-H. Blard[9,11], T. Condom [5], M. Lupker [12], M. Christl[13], Z. He[4], D. Verfaillie[2,14], A. Gorin[7], G. Aumaître[2], D. L. Bourlès[2] & K. Keddadouche [2]

Based on new and published cosmic-ray exposure chronologies, we show that glacier extent in the tropical Andes and the north Atlantic regions (TANAR) varied in-phase on millennial timescales during the Holocene, distinct from other regions. Glaciers experienced an early Holocene maximum extent, followed by a strong mid-Holocene retreat and a re-advance in the late Holocene. We further explore the potential forcing of TANAR glacier variations using transient climate simulations. Since the Atlantic Meridional Overturning Circulation (AMOC) evolution is poorly represented in these transient simulations, we develop a semi-empirical model to estimate the "AMOC-corrected" temperature and precipitation footprint at regional scales. We show that variations in the AMOC strength during the Holocene are consistent with the observed glacier changes. Our findings highlight the need to better constrain past AMOC behavior, as it may be an important driver of TANAR glacier variations during the Holocene, superimposed on other forcing mechanisms.

[1] LGP UMR 8591 CNRS Paris 1 Panthéon-Sorbonne University CNRS, 92195 Meudon, France. [2] Aix-Marseille University, CNRS, IRD, Coll. France, INRAE, CEREGE, 13545 Aix-en-Provence, France. [3] Environnements et Paléoenvironnements Océaniques et Continentaux (EPOC), UMR CNRS 5805, EPOC-OASU Université de Bordeaux, Allée Geoffroy Saint-Hilaire, 33615 Pessac, France. [4] Department of Atmospheric & Environmental Sciences, University at Albany, State University of New York, Albany, NY 12222, USA. [5] Université Grenoble Alpes, CNRS, IRD, G-INP, Institut des Geosciences de l'Environnement (IGE), F38000 Grenoble, France. [6] Department of Earth and Environmental Sciences, Tulane University, 6823 St Charles Ave, New Orleans, LA 70118, USA. [7] Department of Earth & Environmental Sciences Boston College, Chestnut Hill, MA 02467, USA. [8] Climate Change Research Centre, Centre of Excellence, University of New South Wales, Sydney, NSW 2052, Australia. [9] CRPG, UMR 7358, CNRS, Université de Lorraine, 54500 Vandœuvre-lès-Nancy, France. [10] Department of Physical Geography, Faculty of Geosciences, Utrecht University, Utrecht, the Netherlands. [11] Laboratoire de Glaciologie, DGES-IGEOS, Université Libre de Bruxelles, 1050 Bruxelles, Belgium. [12] Department of Earth Sciences, ETH Zürich, 8092 Zürich, Switzerland. [13] Laboratory of Ion Beam Physics, ETH Zürich, 8093 Zürich, Switzerland. [14] Georges Lemaître Centre for Earth and Climate Research, 1348 Louvain-la-Neuve, Belgium. ✉email: jomelli@cerege.fr

Determining the impact of past climate variability on centennial to millennial-scale glacier fluctuations during the Holocene is critical to better understand the respective influence of natural and anthropogenic contributions to current and future glacier retreat[1]. Modern investigations on the glacier–climate relationships only bear on high-frequency forcings due to the short timescale of observations, typically a few decades. Paleoglacial extents, documented from moraine records, offer a unique opportunity to identify key processes controlling millennial-scale glacier behavior. Small (<100 km²) mountain glaciers are particularly relevant for such an analysis[1], due to their short response time.

Summer insolation progressively decreased in the Northern and increased in the Southern Hemisphere since the beginning of the Holocene and is thought to be a major driver for distinct long-term glacier evolution in the two hemispheres[1–4] (Figs. 1 and 2). Consistent with this summer insolation trend, it is generally considered that Northern Hemisphere mid-latitude glaciers increased in extent throughout the mid and late Holocene, recording their maximum size during the Little Ice Age (LIA), while most Southern Hemisphere glaciers gradually retreated until a minimum ice extent was reached during the late Holocene[1,4–8] (Fig. 1). However, direct moraine dating from a growing number of glaciers of the North Atlantic sector[9–12] (i.e. Greenland, Scandinavia, Iceland, and the European Alps) shows larger ice extents during the early Holocene than during the LIA. This observation is inconsistent with the fact that summer insolation has decreased during the Holocene, suggesting other superimposed mechanisms. Moreover, in the tropical Andes, considering both the temperature and precipitation sensitivity of glaciers, large-scale drivers of glacier behavior remain enigmatic. In the Southern tropical Andes, an irregular glacier retreat trend has been observed, despite a progressive increase in austral summer insolation[3]. Moreover, while summer insolation was lower during the mid-Holocene than the late Holocene in the Southern Tropics, no mid-Holocene moraines have been identified using cosmic-ray exposure (CRE) dating[3,13]. This absence of field evidence for a mid-Holocene ice advance may be explained by its obliteration by larger late Holocene advances and suggests the influence of additional forcings superimposed on insolation. New chronologies are thus needed to move beyond specific case studies, possibly reflecting local climate and geomorphological conditions, toward more representative regional and global reconstructions. This will allow for investigations of the possible influence of multiple forcings on millennial timescale glacier changes.

Changes in greenhouse gas concentrations, Northern Hemisphere ice-sheet disintegration, volcanic activity, and land use/cover change, certainly also played an important role in driving climate variations over the Holocene. In addition, the AMOC is known to have a strong influence on Earth's climate, as it plays a critical role in the global oceanic circulation, redistributing heat from the tropics to the Northern Atlantic basin. Recent investigations document strong AMOC variations during the Holocene[14–17] with potential impacts on surface air temperature and precipitation over the continental landmasses surrounding the North Atlantic basin and the tropics. However, to date, the potential influence of AMOC variations on mountain glacier behavior during the Holocene remains poorly constrained[1].

Here we report new in situ ¹⁰Be and ¹⁴C CRE ages from glacial valleys in the tropical Andes (Bolivia), the French Alps, Greenland and Cascade Range (Washington State, USA), along with other published glacier records from the Northern and Southern Hemispheres ("Methods" and Supplementary Data 1 and 2), to document the glacier evolution on millennial timescale during the Holocene. Based on CRE moraine ages, we show that within the

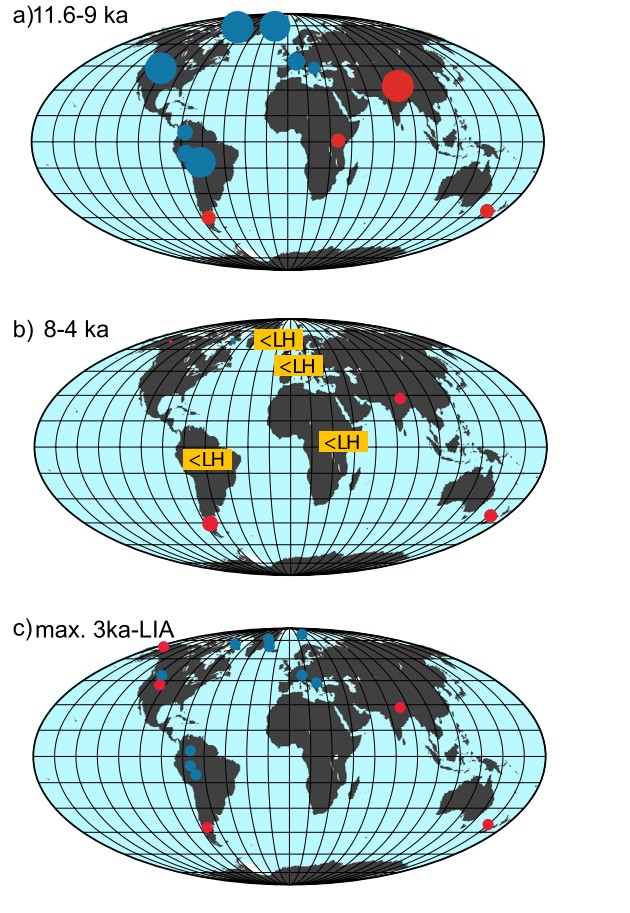

**Fig. 1 Alpine glacier length variations during the Holocene based on new CRE glacier chronologies and those indexed in the ICE-D alpine database.** Mean normalized regional glacier length in different regions and for different time periods, between (**a**) 11.6-9 ka, **b** 8-4 ka, **c** 3 ka-LIA ("Methods" and Supplementary Data 1 and 2). Glacier lengths dated with ¹⁰Be are represented by circles. The sizes of the circles indicate the paleo-ice extent normalized to the LIA extent. Therefore all circles indicate 1 for the LIA and larger circles for other periods indicates a glacial extent larger than during the LIA (1 = LIA extent, 2 = 2 times larger than the LIA). Each symbol corresponds to a mean regional value ("Methods", Supplementary Data 2, n = 66). "<LH" refers to "smaller than the late Holocene" and indicates sites where in situ ¹⁴C and ¹⁰Be from deglaciated bedrock samples or other chronological constraints demonstrate that glaciers were smaller during the documented period than during the late Holocene (Greenland after ref. [12]; Alps after ref. [27]; East Africa after ref. [68], Bolivia, this study). TANAR pattern (blue circles) refers to a larger extent during the early Holocene than during the late Holocene with substantial shrinking during the mid-Holocene.

TANAR, glaciers experienced an in-phase behavior, which differs from other regions of both hemispheres. As this pattern suggests that the North Atlantic and the Tropics were dynamically linked, we explore the possible influence of AMOC variations and its associated large-scale climate impacts, on millennial-scale fluctuations of glaciers in the TANAR. To test the AMOC-influence hypothesis, we compare the TANAR glacier chronologies over the Holocene with recent AMOC reconstructions[14–17] from proxies and transient climate model simulations from LOVECLIM and TraCE[18,19], freshwater hosing sensitivity experiments from five Atmosphere–Ocean General Circulation Models, as well as

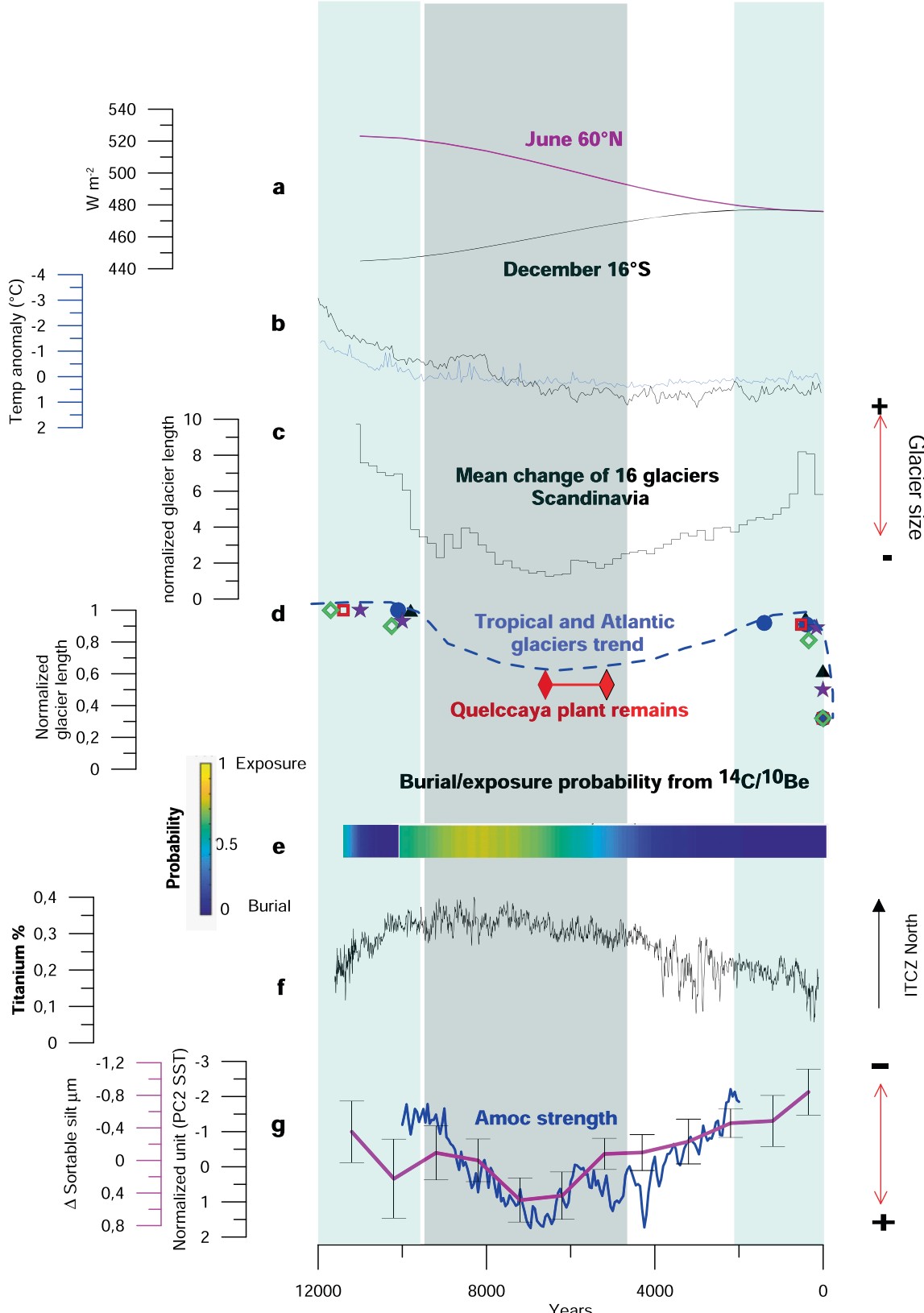

climatic impacts of AMOC variations deduced from a semi-empirical model (see "Methods"). We also investigate the role of other potential drivers (orbital, GHG, ice-sheet extent, volcanic eruptions) through the use of transient Holocene climate simulations including fluctuations in those external radiative forcings.

## Results

**Tropical glacier fluctuations from CRE moraine ages**. To improve our understanding of the regional glacier evolution during the Holocene, we first focus on the tropical Andes. We present new in situ [10]Be and [14]C CRE ages from South and North Charquini glaciers located in the Cordillera Real of Bolivia (16°30'

**Fig. 2 Holocene glacier proxy records in the tropical Andes and in regions surrounding the North Atlantic, compared to external and internal climate drivers. a** Summer insolation changes at 60°N, 16°S, respectively; **b** 30–90°N average annual temperature anomalies from full forcing experiment in TraCE (black, after ref. [18]) and LOVECLIM (blue, after ref. [19]). **c** Holocene glacier activity in Scandinavia (mean of 16 records after ref. [28] (relative glacier size scale). **d** Normalized glacial length extent for different glaciers: green diamond Sisaypampa glacier (Peru, after ref. [26]); blue circle Charquini North glacier (Bolivia, this study); black triangle Charquini South (this study); purple stars Saint-Sorlin glacier (French Alps, this study); blue dot Clavering glacier (Greenland, this study); red square Qori Kalis (Peru, after ref. [69]). The dotted line shows the possible trend of all glacier extents based on plant remains (red diamond after refs. [22,23] showing reduced ice extent conditions between 7 and 5.2 ka at Quelccaya ice cap, Peru) and burial/exposure scenario from (**e**). **e** Probability of burial (blue)/exposure (yellow) from in situ $^{14}C$ and $^{10}Be$ concentrations measured in deglaciated bedrock in front of Charquini North (this study, Bolivia) ("Methods"). **f** Titanium variations from Cariaco after ref. [70]. **g** AMOC changes: blue curve after ref. [14] and purple curve after ref. [15]. Blue (gray) vertical bars show hypothesized large (small) glacial extents from $^{10}Be$ moraine records, in situ $^{14}C/^{10}Be$ modeling and underlying reduced (enhanced) AMOC strength, respectively.

S; 68°10' W) (Fig. 1 and Supplementary Figs. 1 and 2). On Charquini South, $^{10}Be$ CRE ages from rock samples collected from the outer LIA moraine, previously dated by lichenometry[20], confirm a LIA age with a mean moraine age of $420 \pm 15$ a ($n = 7$) (Supplementary Data 1, Supplementary Fig. S1, and "Methods"). In addition, the nearest downslope moraine was formed during the early Holocene ($10.0 \pm 0.21$ ka; $n = 4$). On Charquini North, distal to the LIA moraine, the next moraines were dated to $1.20 \pm 0.12$ ka ($n = 3$) and to $10.10 \pm 0.26$ ka ($n = 4$), respectively (Supplementary Data 1, Supplementary Fig. S2, and "Methods"). The absence of moraines from the mid-Holocene suggests that glaciers were smaller than their late Holocene extent. A retreat-advance scenario modeled from combined in situ $^{10}Be$ and $^{14}C$ concentrations ("Methods") in deglaciated bedrock located in-between the current front position and the late Holocene moraines on Charquini North independently reveals that during the mid-Holocene the front of the glacier was positioned up-valley from its LIA extent (Supplementary Fig. S2 and "Methods"). The combination of moraine and bedrock chronologies with independent data from lake sediment records and plant remains[21–25] and a few other moraine records[13,25,26] suggests that at the end of the deglaciation, that started about 19,000 years ago, glaciers advanced or stabilized during the early Holocene for a period long enough to form moraines that correspond to the maximum Holocene extent before retreating. During the mid-Holocene, glaciers remained in the upper part of the valleys, followed by an advance starting at ~4–2 ka, ultimately culminating in the late Holocene, either during or a few centuries prior to the LIA (Fig. 2 and Supplementary Figs. S1 and S2).

**In-phase millennial-scale glacier chronologies in the North Atlantic region**. We compare Andean glacier chronologies with new and published moraine records from other small glaciers in the North Atlantic regions, such as the Alps and northeastern Greenland (Fig. 1, Supplementary Figs. S3–S5, Supplementary Data 1, 2, and "Methods"). The millennial-scale evolution of tropical Andean glaciers during the Holocene mimics glacier fluctuations in these regions, although the timing of individual centennial-scale moraine deposition differs. Our new $^{10}Be$ chronology from the proglacial margin of the French Saint-Sorlin glacier (45°N; 6°17' E) reveals a LIA glacial extent close to the maximum Holocene ice extent that occurred during the early Holocene (Supplementary Data 1 and Supplementary Fig. S3), in agreement with other glaciers in the European Alps[9,10] ("Methods" and Supplementary Data 2). Moreover, several previous studies[1,27] provide evidence of reduced glacier extents during the mid-Holocene in the Alps. At higher latitudes, recent CRE moraine ages from a glacier adjacent to Clavering ice cap (East Greenland, Supplementary Data 1, Supplementary Fig. S4) reveal a similar pattern of changes, in line with records from other glaciers in Greenland[11,12], in Scandinavia, and in Iceland[28,29] (Figs. 1 and 2 and Supplementary Data 2). Altogether, CRE

chronologies attest to a common Holocene glacier pattern in the TANAR, showing a maximum Holocene extent during the early Holocene, followed by a strong retreat during the mid-Holocene and a re-advance in the late Holocene, culminating during or prior to the LIA (Figs. 1 and 2). In other regions of the Northern Hemisphere, such as the Western USA or Himalayas, mountain glacier chronologies suggest less uniformity and differences with the TANAR glaciers. While most glacier fluctuations in the Cascade Range (Washington State, USA) (Supplementary Fig. S5) mimic the TANAR glaciers, in Alaska the maximum Holocene extent did not occur during the early Holocene but during the late Holocene. The glacier records in the Himalayas exhibit a complex pattern (Fig. 1), depending on the relative influence of precipitation related to the position of the westerlies or to monsoon intensities[1] on their mass balance. In the mid-latitudes of the Southern Hemisphere, glaciers generally recorded a progressive ice retreat following the early Holocene which corresponds to the maximum Holocene ice extent[4–7].

**Glacier changes in the TANAR and associated forcings**. Climate evolution during the Holocene was investigated using transient Holocene experiments from TraCE[18] and LOVECLIM[19] models including all available external forcing at the time they were performed. The TANAR glacier evolution is not fully consistent with these transient Holocene simulations (Fig. 2), suggesting other processes than external forcings (orbital, GHG, ice-sheet extent, volcanic eruptions) might have impacted their evolution. The LOVECLIM experiment, which includes a full transient forcing, simulates a rather monotonic temperature change in the Northern mid-latitudes that does not fit the glacier pattern presented above (Fig. 2). In the tropical Andes, the full forcing LOVECLIM experiment does not suggest any significant change in precipitation during the Holocene and a modest increase in temperature, inconsistent with our moraine record (Supplementary Fig. S6). To further explore the influence of possible forcing on glacier fluctuations we investigate internal climate variability drivers, such as El Niño-Southern Oscillation (ENSO). This is one of the largest sources of internal climate variability with strong impacts on current tropical glacier behavior; hence it is a potential candidate to explain the non-monotonic Holocene glacier fluctuations in the Andes[30,31] ("Methods"). Current observations show that El Niño events induce a very negative mass balance of tropical Andean glaciers and associated retreating trends[2,32,33]. On the other hand, La Niña conditions are generally associated with cooler and wetter conditions at the glacier surface causing positive mass balances[33]. However, recent investigations of both variance and frequency of paleo-ENSO proxies during the Holocene[31,34] suggest El Niño-like conditions during the early Holocene and LIA[30,31,34], when major glacier advances occurred. Moreover, this discrepancy between ENSO and glacier records is also apparent during the mid-Holocene. While CRE ages, plant remains and sediment records show a limited ice extent during

the mid-Holocene, Niña-like conditions are generally documented during this period[30,34]. This disagreement suggests that long-term glacier change in the Andes might not be directly controlled by ENSO variability.

## Discussion

As millennial-scale AMOC changes influence both temperature and precipitation, we test how AMOC changes could influence the TANAR glacier evolution during the Holocene. In the high- and mid-latitudes of the Northern Hemisphere, the LOVECLIM and TraCE model experiments reveal that during the early Holocene, temperatures were mostly driven by ice-sheet remnants and AMOC-related changes[19,35] (Supplementary Figs. S6 and S7). Recent AMOC reconstructions[14–17] also suggest that the AMOC may have influenced precipitation in the tropical Andes throughout the Holocene, thereby affecting glacier evolution (Fig. 2) ("Methods"). We propose that AMOC changes also significantly impacted temperature during the mid- and late Holocene in the mid- and high latitudes of the Northern Hemisphere, superimposed on summer insolation that reached a minimum during the last millennium. To support this hypothesis, we assess and explore a potential AMOC climatic fingerprint by first correlating global sea surface temperatures (SST) against the recent AMOC reconstruction[17] over the instrumental era (Fig. 3a, b). An enhanced AMOC is characterized by a warming of the North and tropical Atlantic and a cooling in the South Atlantic. This results in an Atlantic SST interhemispheric seesaw pattern associated with a strong warming signal in the Northern tropical Atlantic region. Such an AMOC fingerprint is consistent with numerous climate model simulations[36–38]. Secondly, we show that North Atlantic SST impacted by AMOC variations are also significantly anti-correlated with tropical Andean glacier mass balance at decadal timescales (Fig. 3c). This relationship between North Atlantic SST and tropical Andean glacier mass balance hinges on the migration of the Intertropical Convergence Zone (ITCZ). Those changes can be both driven by insolation and AMOC changes. Indeed, insolation-driven low North Atlantic SST would displace the ITCZ to the south and lead to an intensification of the South American summer monsoon (SASM). The resultant enhanced moisture influx and convergence over tropical South America[39,40] would lead to a positive glacier mass balance. As the ITCZ position can also be modulated by the AMOC strength and the associated interhemispheric SST dipole[41], we deduce from these teleconnections that AMOC variations also have the potential to impact the TANAR glacier fluctuations in unison on multi-centennial timescales.

We thus analyzed whether the TANAR glacier fluctuations are consistent with Northern Atlantic SSTs and AMOC changes throughout the Holocene using AMOC reconstructions produced by recent independent proxy records[14–17,42] ("Methods") (Figs. 2g, 3 and 4). These reconstructions are mainly consistent and reveal two minima during the early and late Holocene, including the LIA, interrupted by a phase of enhanced AMOC between 8 and 4 ka, with the strongest AMOC reconstructed at around 7 ka. This AMOC maximum is in-phase with a small extent of the TANAR glaciers, thus suggesting that AMOC strengthening could have served as a potential driver of the mid-Holocene thermal maximum in the Northern Atlantic regions and drier conditions in the tropical Andes[40], enhancing glacier retreat (Fig. 4 and Supplementary Figs. S6 and S7). Conversely, an AMOC weakening associated with reduced northward oceanic heat transport would support Northern Atlantic cooling and, hence, glacier advance during the early and late Holocene. It also shifts the ITCZ position southward, enhancing related precipitation over the tropical Andes and cooler temperatures due to

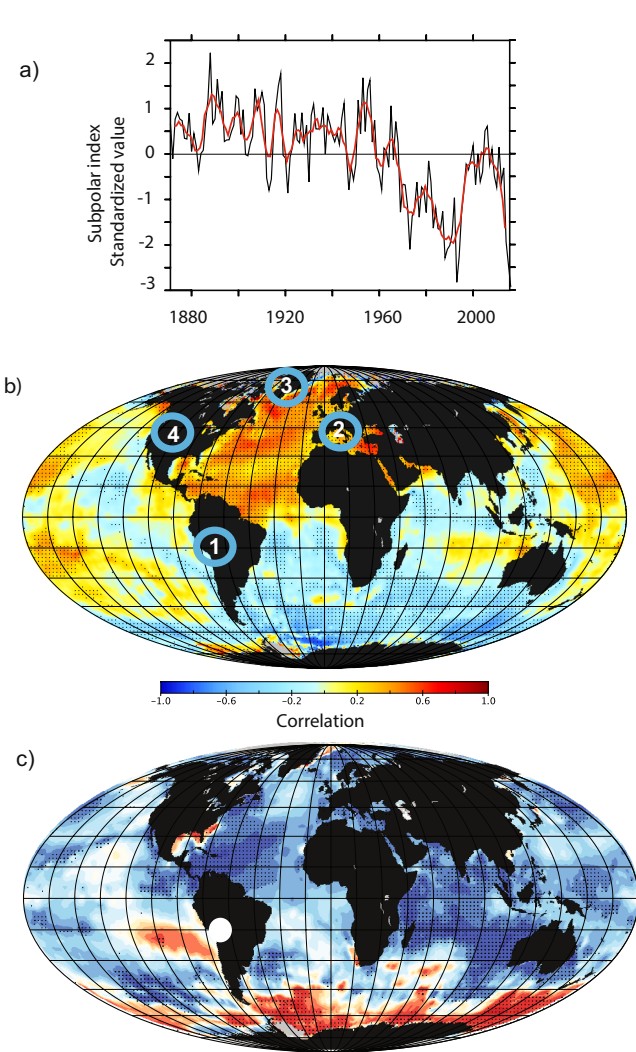

**Fig. 3 Global sea surface temperatures (SST) related to the subpolar index and glacier records. a** Subpolar index from ref. [17]. In black is the annual mean and in red is the 10-year running mean. **b** Correlation between detrended SST and HadISST on the Subpolar index shown in panel **a** that might represent AMOC variations over the same period. Stippling indicates a significant correlation coefficient, i.e. different from zero following a Student t test at the 95% level. Blue circles show locations with new CRE chronologies (see "Methods"), (1) Charquini North and South glaciers (Bolivia), (2) St Sorlin glacier (French Alps), (3) clavering glacier (Greenland), (4) enchantment lakes (Cascade Range, US). **c** Spatial correlation between October–March mass balance on Bolivian Zongo glacier[33] (white dot) and global SSTs over period 1974–2016. Both mass balance and SST data were smoothed with a 9-year running mean (see "Methods").

positive land-surface feedbacks[13] that favor large tropical glacier extents in the early and late Holocene, thus producing the overall TANAR signature.

We notice, however, a strong discrepancy between the recent AMOC reconstructions and the one simulated in the transient numerical experiments (Supplementary Fig. S6). While the LOVECLIM and TraCE, transient numerical simulations show a minimum around ~8 ka, followed by a steadily increasing simulated AMOC strength through the rest of the Holocene, the reconstructed AMOC shows a similar low during the early Holocene but with a maximum in the mid-Holocene, followed by

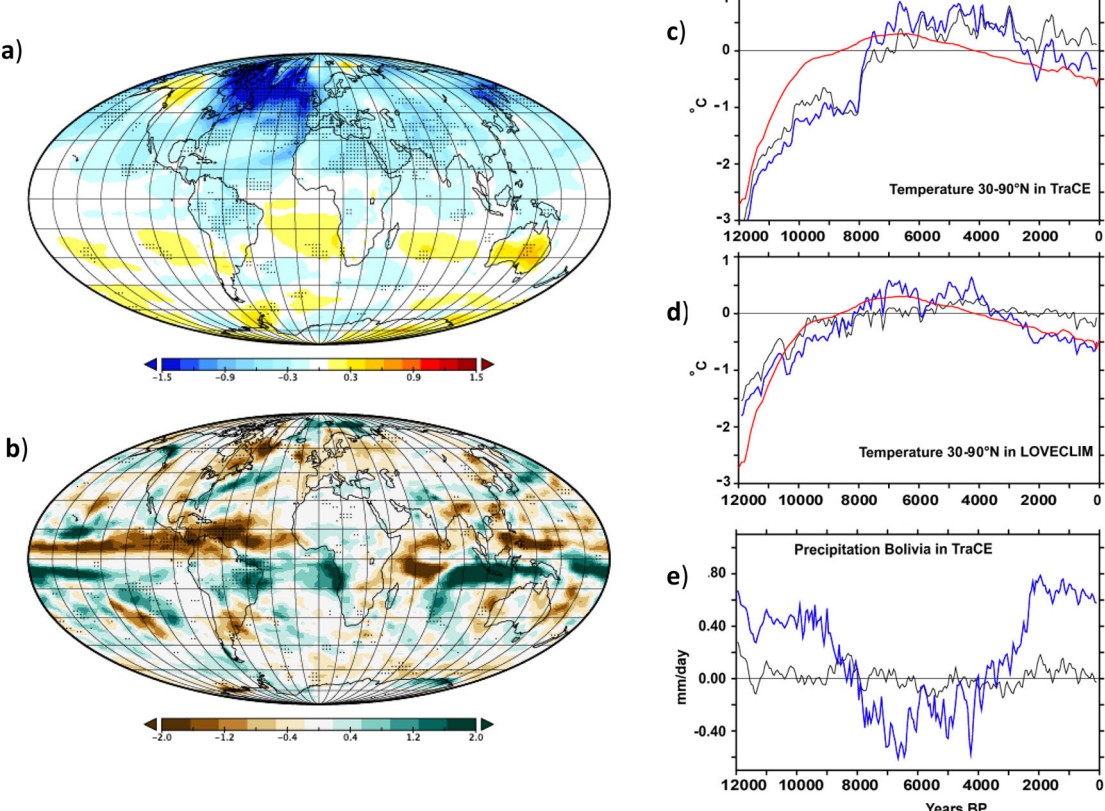

**Fig. 4 Temperature and precipitation changes related to AMOC changes. a** Ensemble mean surface temperature change (annual mean value in °C) of five models, showing the difference between hosing and control simulations (blue color shows colder temperatures compared to the control simulation and green color higher precipitations). The stippling indicates where all the model simulations exhibit the same sign of changes. This pattern of temperature change is mainly related to reduced AMOC strength. **b** Same as (**a**) but for precipitation (mm/d). **c**, **d** 30–90°N AMOC adjusted Holocene temperature changes (blue curves) based on the semi-empirical linear model for TraCE simulation and LOVECLIM simulation (black curves) respectively as compared with temperature reconstruction from ref. [45] (red curve). The standard deviation used is 1.2 Sv for the AMOC. **e** same as (**c**, **d**) but for precipitation in the Charquini region (tropical Andes of Bolivia 16°S, 68°W). A 100-year running mean has been applied to the time series in (**c–e**).

a weakening towards the late Holocene[43] (Supplementary Fig. S6). To better quantify the effects of a weak AMOC on regional climatic conditions during the early and late Holocene, freshwater hosing experiments are used from five Atmosphere–Ocean General Circulation Model (AOGCM) simulations[44] ("Methods") to guarantee the robustness of the simulated signals, notably at the regional scale. Results show that a weak AMOC is associated with an atmospheric temperature decrease over the TANAR. The weak AMOC also leads to a more southerly location of the ITCZ, thereby increasing precipitation in the Southern tropical Andes through an enhancement of the South American monsoon[39], in agreement with proxy records (Figs. 2 and 4a, b, e and Supplementary Fig. S6).

To go one step further, we develop a semi-empirical model, merging the simulated response of temperature and precipitation to external forcing in both transient paleoclimate simulations, with impacts of AMOC variations over the Holocene, based on the three recent AMOC reconstructions[14–17,42] and AMOC fingerprints from the five hosing sensitivity tests from AOGCMs mentioned above ("Methods"). For this purpose, we calibrate the amplitude of AMOC changes using a hemispheric-scale reconstruction of temperature[45] ("Methods"). From this calibrated AMOC reconstruction, we develop corrected simulated temperature and precipitation changes in TANAR (Fig. 4c–e and Supplementary Fig. S6). Results clearly highlight that the inclusion of AMOC variations on top of the simulated response to external forcing produces a climate response consistent with

glacier changes in the TANAR. While AMOC-corrected temperature changes remain modest in the Northern Hemisphere, this forcing still causes cooler temperatures over the late Holocene, corresponding to glacier advances. In the tropical Andes, AMOC-corrected precipitation changes show large variations in agreement with independent reconstructions[8,21,24] which is not the case in the full forcing experiments (TraCE and LOVECLIM) that do not include the effect of the reconstructed AMOC variations. This suggests a potential influence of AMOC variations on regional climate conditions in the tropical Andes during the Holocene. The maximum AMOC change quantified through this method is ~4–5 Sv over the whole Holocene (Supplementary Fig. S6). The difference in AMOC strength between the mid-Holocene (6 ka) and the preindustrial period is 3–4 Sv, which is in agreement with several PMIP3 models[43], providing further confidence in the result obtained from the calibration using the recent 30–90°N temperature reconstruction[45].

From Fig. 4 and our conceptual model results, we can estimate that a weak AMOC would likely cause a ~0.5 °C temperature decrease north of 30°N and a ~1 mm/day precipitation increase (~40% increase compared with today) in the tropical Andes between the mid-Holocene (mean over 6–5 ka) and the last millennium (mean over 1–0 ka). In boreal summer the AMOC may account for up to 1/3 of the cooling seen during the late Holocene in the Northern Hemisphere, while the other 2/3 are primarily the result of insolation. In the tropical Andes, the AMOC clearly overwhelms the impact of radiative forcing from

insolation and greenhouse gases as estimated in transient simulations, which suggests that the AMOC might have been a potential driver of glacier evolution over large parts of the Holocene in the tropical Andes. It also potentially may have played a secondary but still significant role in the mid-latitude of the Northern Hemisphere as highlighted in Fig. 4.

We stress that more sophisticated approaches would be needed to obtain a more robust quantification of AMOC-induced climate changes at the glacier surface. However, our simple semi-empirical model reveals the potential influence of AMOC variations on the simulated temperature and precipitation in the TANAR from transient simulations (Fig. 4). The strong disagreement between AMOC variations seen in proxy records and those produced by transient model simulations highlights the need for a better understanding of the causes of AMOC variations over the Holocene. Yet the drivers of the AMOC dynamics for this period are still poorly understood. AMOC changes may be related to external forcings, meltwater input[43], or other missing processes, such as changes in Mediterranean outflow during the so-called Sapropel S1 event[46]. In addition, following the "Zealandia switch" hypothesis[47], a poleward shift of the SH westerlies between the early- and mid-Holocene would have led to global warming. We note that this poleward shift of the SH westerlies could have also led to an AMOC strengthening, partly through the modulation of the Aghulas leakage, in agreement with our suggested AMOC evolution through the Holocene.

New transient simulations using higher resolution climate models and further proxy comparisons appear necessary to make progress on this attribution issue. This is of primary importance as glaciers in the TANAR show a strong retreat rate today, despite potential AMOC weakening. Indeed, recent estimates of $3 \pm 1$ Sv weakening of the AMOC[17] over the last century are comparable to the amplitude of estimated changes over the last 6 ka and could be partly driven by internal variability. This weakening, in the case of our simple conceptual model ("Methods"), translates to a potential cooling of about 0.5 °C north of 30°N compared to preindustrial values, which acts as a negative feedback on the current glacier melting. Nevertheless, the regional effects of AMOC weakening are clearly overwhelmed by global warming induced by radiative forcing from increased greenhouse gas concentrations induced by human activities[48]. CMIP5 projections do not account for significant ice mass loss from Greenland, a mechanism that could lead to a larger AMOC decrease than projected[49]. The IPCC Special Report on Ocean and Cryosphere in a Changing Climate assesses an AMOC intensity decrease of $-32 \pm 14\%$ for RCP 8.5 scenarios by 2100[48,50]. Since then, CMIP6 models still poorly account for Greenland ice-sheet melting, but do show slightly stronger sensitivity in the different socioeconomic development (ssp) scenarios, with an estimate of 34–45% weakening of AMOC present-day strength at the end of this century in different ssp scenarios.

Such a weakening of the AMOC may induce, by itself and not accounting for radiative forcing from greenhouse gases, a potential regional cooling of ~2 °C north of 30°N as compared to preindustrial values. A future weakening of the AMOC may thus dampen regional climatic changes north of 30°N, with a temperature reduction that could amount to 30–40% of the absolute temperature increase estimated by some CMIP5 models for 2100 in a high-emission scenario[48,50]. Given that the AMOC weakening might also affect tropical precipitation going forward, it constitutes a large source of uncertainty in future climate scenarios. Better understanding past AMOC dynamics, notably over the Holocene, when the climate mean state was close to present-day, is therefore crucial for our understanding of future climate change and being able to implement better adaptation strategies.

## Methods

We report new CRE chronologies of five small mountain glaciers. We assume that moraines were formed by the glacier at times when it was in (or close to) equilibrium with the climate. The moraine age corresponds to the end of the glacier advance or still stand. The glacier may have retreated between two successive moraine depositions observed in the field, but the front never surpassed the downslope ridge.

### In situ [10]Be and [14]C and laboratory analysis.
All moraine samples except for those from Charquini South (CQ13-1 to 6) and Enchantment lakes were processed for [10]Be extraction at CALM lab (Cosmonucléides Au Laboratoire de Meudon, France) and LN2C (CEREGE, France) following chemical procedures adapted from[51]. [10]Be/[9]Be ratio measurements were performed at the ASTER AMS national facility (CEREGE, Aix-en-Provence) using STD11 standard[52]. Charquini South moraine samples CQI 13-1 to -6 were processed at the Geological Institute, ETH Zürich and measured on the TANDY AMS at ETH Zürich[53], and data were normalized to the ETH in-house standard S2010N ([10]Be/[9]Be = $3.3 \times 10^{-12}$) calibrated against ICN 01-5-1[54]. Enchantment lakes [10]Be moraine samples were processed in the Lamont-Doherty Earth Observatory Cosmogenic Nuclide Lab and Be isotope ratios measured at the Center for Accelerator Mass Spectrometry at Lawrence Livermore National Laboratory.

Extractions of in situ [14]C and [10]Be from the bedrock collected at Charquini North glacier were carried out in the Tulane University cosmogenic-nuclide laboratory. The in situ [14]C extraction method is presented in ref. [55]. [14]C/[13]C ratios were measured at the Woods Hole Oceanographic Institution - National Ocean Sciences Accelerator Mass Spectrometry facility relative to the NIST SRM 4990C Oxalic acid-II standard, prepared in the same graphite reactors used to prepare the samples, to maintain complete internal standardization. Stable carbon isotope ratios are measured from a small (~2 μg C) aliquot at the University of California-Davis Stable Isotope Facility.

All [10]Be/[9]Be isotope ratios are reported following the 07KNSTD standardization[54] and new [10]Be moraine ages were determined using the CREp online calculator[56] with the LSD scaling model[57], assuming no denudation and using local production rates from ref. [58] in the tropical Andes, and the mean Arctic-NE rate of $4.05 \pm 0.23$ at/g/year in Greenland and in the French Alps. Throughout the paper, tables and figures, we report individual CRE ages with their associated uncertainties that include the standard deviations of both analytical (reported in Supplementary Data 1) and production rate uncertainties. Published chronologies were not recalculated as they were already based on recent productions rates. For a given moraine, its age corresponds to the weighted mean of the sample ages (calculated with their analytic uncertainty), after discarding outliers that were identified with a Chi-square test[59].

### [14]C/[10]Be numerical model.
Whereas moraine boulders typically experience a simple history with a single period of exposure and negligible erosion following deposition, proglacial bedrock samples may have undergone multiple episodes of exposure, burial by ice, and subglacial erosion. Therefore, we used a numerical model that simulates nuclide production, decay, and erosion to test possible glacier histories that yield cosmogenic-nuclide concentrations and ratios in agreement with the measured values for the Charquini North bedrock samples P12 and P13. The model simulates a bedrock depth profile of [10]Be and [14]C concentrations through time for various exposure scenarios, driving production when exposed, and decay and glacial erosion when ice-covered. Production and decay rates are fixed, but erosion rates are systematically adjusted to explore a range of possibilities for each exposure scenario. Production rate profiles are derived from code based on the CRONUS online calculator v3 using the LSDn scaling scheme[57]. [14]C production rates are based on the long-term average value of CRONUS-A measured at Tulane University[55]. The evolution of nuclide concentrations in a bedrock column is then driven by a prescribed exposure/erosion scenario in 100-year time steps via Eq. (1).

$$N(z,t) = P_{NT}(z) \cdot t + N(z, t-1) \cdot e^{(-\lambda_N \cdot t)} \tag{1}$$

Where $N$ is the concentration of the nuclide in the bedrock as a function of depth ($z$) and time ($t$), $P_{NT}$ is the total production of the nuclide via spallation and muon production as a function of depth, and $\lambda_N$ is the decay constant of the nuclide. During times of exposure, the model uses the production portion of Eq. (1) (left of the addition sign). During times of burial, the model only uses the decay portion of Eq. (1) (right of the addition sign). Erosion is incorporated by redefining the "surface" at some depth below the top of the bedrock. The model assumes that erosion only takes place during times of burial. 100,000 unique exposure/burial scenarios were randomly generated with the intent to sample a wide range of Holocene histories. For a given scenario, each time step was assigned exposure or burial based on a specified probability ($P$) that it would match the designation (exposure or burial) of the previous time step. For example, in a scenario with $P = 0.8$, each time step has an 80% probability of having the same designation (exposure or burial) as the previous time step. $P$ ranged from 0.6 to 0.99 across the 100,000 scenarios to ensure that exposure and burial intervals fluctuate over a wide range of frequencies (e.g., scenarios with $P = 0.6$ tend to have more high-frequency variability, scenarios with $P = 0.99$ tend to have more extended intervals of exposure and burial). The down-valley moraine ages were used to constrain the

exposure scenarios; only scenarios with burial prior to $10.10 \pm 0.26$ ka as well as during the $1.20 \pm 0.12$ ka interval were considered. The model tested erosion rates of 0 to 50 mm/ka (in steps of 1 mm/ka) with each scenario (Supplementary Fig. S8).

Each sample was stepped through each exposure scenario (testing all erosion rates) individually, and the scenario was saved if the final surface $^{14}C$ and $^{10}Be$ concentrations were within $3\sigma$ uncertainty of the sample concentration, including both measurement and production rate (7.3% for $^{14}C$ and 8.3% for $^{10}Be$, $1\sigma$; ref. [60]) uncertainties, added in quadrature. Scenarios that successfully simulate both samples were considered plausible Holocene histories.

**Fluctuations of Charquini North.** Six recently deglaciated bedrock samples were collected from two areas in the forefield of Charquini North. For all samples we assume that (1) subglacial erosion before 17–11.6 ka removed all cosmogenic concentrations from previous exposure; (2) subglacial nuclide production from penetration of cosmic radiation through the ice is negligible; this potentially concerns muogenic $^{14}C$ production in particular; (3) the samples at each of the two areas had the same millennial exposure/burial history.

The upper bedrock area, where samples P1, P2, P3, and P3b were taken, is located a few meters downslope of the current terminus. These samples were collected on a gentle rock surface located at a distance of about 150 m from the summit (see Supplementary Fig. S2). The lower bedrock area, where P12 and P13 were sampled, is located at 900 m from the terminus, halfway between the current terminus and the LIA moraines[61,62].

Samples P1 and P2 became ice-free during the last decade based on aerial photos in 2013, while P3 and P3b were already ice-free in the 1970s. These four samples have very low $^{10}Be$ and $^{14}C$ concentrations corresponding to 50–100 years ($^{10}Be$) and 170–250 years ($^{14}C$) of apparent exposure. Two general exposure/burial scenarios are possible to explain the low nuclide concentrations. First, the samples were almost constantly covered by the glacier throughout the Holocene, preventing the accumulation of higher nuclide concentrations at/near the surface. Second, the samples were ice-free for extended periods within the Holocene, leading to the production of higher nuclide concentrations, but subglacial erosion during subsequent glacier advances substantially reduced the higher nuclide concentrations.

The two samples P12 and P13 have significantly higher $^{10}Be$ and $^{14}C$ concentrations than the four samples further upslope corresponding to apparent (minimum) exposure durations of $2.99 \pm 0.15$ ka and $3.61 \pm 0.22$ ka, ($^{10}Be$) and $0.79 \pm 0.05$ ka and $1.08 \pm 0.06$ ka ($^{14}C$), respectively (Supplementary Data 1). The $^{14}C$ concentrations are depleted compared to the $^{10}Be$ concentrations, when assuming continuous exposure without burial. This discrepancy combined with the downslope moraine evidence of recent glacier cover imply that these surfaces were ice-free for a certain period during the Holocene, and that erosion during subsequent glacier cover did not remove all nuclides accumulated during exposure[55,60,63]. Based on the above-described numerical modeling, the combined $^{10}Be$ and in situ $^{14}C$ concentrations indicate that these sample spots were most probably ice-free between ~9.5 and 5 ka and mostly covered by ice from 4 ka onward (Fig. 2g). Subglacial erosion rates linked to this scenario are ~0–30 mm/kyr. This exposure/burial scenario agrees with the moraine record in the catchment, which reveals at least three periods of glacier advance during the Holocene. The first occurred during the early Holocene, dated to $10.10 \pm 0.26$ $^{10}Be$ ka based on four samples (CQ11-4, Cbn 11, 13, and 15). The two other advances, smaller in amplitude, occurred during the late Holocene (mean $^{10}Be$ age of $1.20 \pm 0.12$ ka) and the LIA[61,62].

**Other glacier fluctuations.** On Charquini South glacier (Supplementary Fig. S1), samples Cb12 ($0.22 \pm 0.05$ ka) and R2 ($0.23 \pm 0.05$ ka) collected on the LIA moraine, and CQ13-3 from the early Holocene moraine dated to $12.38 \pm 0.55$ ka, were rejected based on a $Chi^2$ test (based on analytic uncertainties) and considered as outliers (Supplementary Data 1).

On Saint-Sorlin glacier (French Alps 45°18' N; 6°17' E; Supplementary Fig. S3 and Supplementary Data 1) the first moraine downslope from the historically documented LIA[64] yields a mean $^{10}Be$ age of $9.79 \pm 0.49$ ka (S1, S2, S4 $n = 3$); the following downslope moraine gives a mean $^{10}Be$ age of $11.40 \pm 0.44$ ka ($n = 5$; S6–10). Distal from these two early Holocene moraines, two other ages collected on a lateral moraine remain located below the highest lake (see Supplementary Fig. S3) yield a mean age of $11.20 \pm 0.58$ ka.

In Northeastern Greenland, we focused on a small glacier on Clavering (74°37' N; 21°33' W) where eight $^{10}Be$ samples were collected on moraine remains (Supplementary Fig. S4 and Supplementary Data 1). Two end moraines are visible close to the current terminus position. The closest moraine from the current terminus was probably formed during the last centuries based on aerial photographs and a single sample (Zack 41) dated to $0.28 \pm 0.25$ ka. Downslope, three samples collected on another end moraine give a mean $^{10}Be$ age of $1.17 \pm 0.09$ ka (Cla 51, 52, and 53). At ~1 km distance from the current terminus position, a sample collected on an end moraine was dated to $9.55 \pm 0.96$ ka (Cla 49). The small lateral moraine where Cla 50 was collected and dated to $10.53 \pm 0.68$ ka may correspond to the same glacial landform. Grouping Cla 49 with Cla 50 gives a mean $^{10}Be$ age of $10.20 \pm 0.55$ ka. Two other samples (Cla 45–46) located at ~1.4 km from the current terminus on another moraine yield a mean $^{10}Be$ age of $10.10 \pm 0.49$ ka.

Grouping the samples from these two moraines (Cla 45, 46, 49, 50) gives a mean age of $10.15 \pm 0.37$ ka.

Samples from the Enchantment Lakes Region, central Cascades, Washington, USA, are presented demonstrating the Holocene response. The Enchantment Lakes moraines and associated samples were not deposited by a single glacier, but rather a set of coalescing glaciers that left behind composite moraine complexes. One set has previously been dated to the earliest Holocene[65], likely during the slowdown in retreat from the LGM maximum and an inner moraine set. Here we focus on the late Holocene. We take a conservative approach to interpret the results for the inner moraine complex and infer two main advances during the late Holocene, one corresponding to the Neoglacial (e.g., 4–2 ka) and the other generally with the Little Ice Age.

To stress the relevance of our glacier records, our dataset was compared to other already published moraine chronologies ($n = 66$). We focused on small glaciers with moraine chronologies based on CRE ages given in the worldwide Ice-D database (http://alpine.ice-d.org/), except the reconstruction from ref. [28], based on a compilation of 16 glaciers documented from lake sediments and located in the North Atlantic sector. Four criteria were considered:

1. We did not consider catchments with moraines that may correspond to the early Holocene or an older period such as the Younger Dryas if uncertainties are too large to distinguish between YD and Early Holocene.
2. We did not select catchments with undated moraines in-between the LIA and late glacial moraines.
3. We did not take into account landforms with only one CRE age or unpublished data.
4. Poorly constrained moraines such as Mt Conness Holocene moraines (http://alpine.ice-d.org/site/CONA) with a large dispersion in the dataset were not considered.

Consequently, we distinguished three categories, glacier that fit with the AMOC changes, those that do not and finally glaciers for which their change is ambiguous. This ambiguity is related to the following possibilities:

a. The evolution of the glacier during the early Holocene is not analyzed
b. The chronology of the early Holocene remains uncertain and may attribute either to the Younger Dryas or during the mid-Holocene due to dating uncertainties.

Glaciers of the two first categories (fit with TANAR pattern or does not fit) were considered in our analysis while those belonging to the third category were not.

Lake sediment records that were combined with CRE ages by the authors were preferentially selected for our comparative analysis. We noticed some differences in the individual age of moraines between glaciers. These differences may be related to local climate conditions, different response times and other forcings that are not explored in this paper. Here, we focused on the major forcing responsible for glacier changes during the onset-early Holocene, mid-Holocene, and late Holocene (including the LIA).

The glacier length in Fig. 1 is considered as the distance between the terminus position in a given period and the current position. This length is normalized for each glacier by dividing the glacier length at any period by the maximum length obtained during the late Holocene based on moraine records. This allows computing length values of 1 during the LIA for most glaciers. During the early Holocene, glaciers were larger than at any time thereafter in the TANAR. These early Holocene moraines in the TANAR reveal that during the deglaciation, which started earlier, at least one still stand and even an advance occurred in the early Holocene, long enough to form the investigated moraines in this paper. During the mid-Holocene, values exceeding 1 were present mostly in the Southern Hemisphere. In the Northern Hemisphere and in the tropics glaciers were generally smaller than during the late Holocene, except in Asia. More data are needed to better constrain glacier fluctuations in East Africa as only one glacier is referenced there.

**Links between temperature/precipitation and tropical mass balance.** At the global scale, temperature and precipitation changes may both significantly impact glaciers. In extratropical regions, temperatures are generally well correlated with some variables of the energy budget, and it is generally assumed that temperature changes largely controlled the long-term glacier behavior during the Holocene. However, in the outer tropics, where Zongo glacier is located (i.e. the glacier used in this paper as a target to investigate relationships between climate conditions and mass balance; see Fig. 3 for instance), it is now well established that current glacier mass balance is mainly controlled by precipitation changes[2]. For instance, direct observation on Zongo glacier (Bolivia) show that about 70% of mass balance variability is controlled by precipitation between September and March, whereas ablation and accumulation are limited during the rest of the year. In addition, several papers conducted on direct mass and energy balance measurements show that there is almost no relationship between temperature and melting at a seasonal and annual timescale in the outer tropics. This is mainly due to the fact that there are only limited variations of temperature in the tropics at seasonal or annual timescale (see for instance, refs. [2,33]). However, at an orbital timescale, we assumed that temperature may have a minor influence on tropical glacier changes as identified in ref. [13]. Since glaciers in the extratropical and in tropical region

respond differently to precipitation and temperature changes, we considered both variables in our analysis. We are showing that the AMOC had an influence on both variables.

**Links between Pacific oscillations and moraine records**. Pacific Decadal Oscillation (PDO) changes have been reconstructed from various proxies. The longest one[66] revealed cold tropical Pacific SSTs during the Medieval Climate Anomaly (MCA), and warm SSTs during the LIA. Observations of tropical Andean glaciers conducted over the last decades show a positive correlation between mass balance and cold SSTs and vice versa (Fig. 3). This recent relationship is inconsistent with reconstructed PDO changes during the MCA-LIA.

**Model simulations**. Two transient simulations of the Holocene based on the TraCE, using the CCSM3 model[18], and LOVECLIM model[19,67] are used to evaluate whether external forcings can explain our glacier records. The TraCE simulation includes variations in insolation, greenhouse gas concentrations, ice-sheet extension, and altitude from the ICE5-G reconstruction and also a crude estimate of associated freshwater release in the North Atlantic. LOVECLIM uses similar forcings, but ICE-6G for the ice sheet, and does not account for freshwater release in the North Atlantic. This second transient simulation also includes an estimate of volcanic forcing, which was neglected in the TraCE simulation.

**AMOC reconstructions**

*Instrumental era*. Using SST variations in the North Atlantic subpolar gyre (SPG)[16], we performed a regression on detrended SST from HadISST with the associated AMOC index over the instrumental era (Fig. 3). We show that positive fluctuations in the SPG, corresponding to AMOC enhancement, are related with a large-scale bipolar seesaw in atmospheric temperature, reminiscent of the AMOC signature in freshwater hosing simulations[14,15].

In addition, links between tropical glacier mass balance and SSTs over the recent decades were explored through the longest seasonal mass balance time series in the tropical Andes from the Bolivian Zongo glacier, located close to Charquini glaciers in Bolivia[33]. As shown in Fig. 3, a positive mass balance of Zongo glacier is associated with cold SSTs in the Northern Atlantic region.

*Holocene*. The three AMOC reconstructions used in this paper have not been calibrated with observed AMOC changes because this circulation is only measured directly since 2004. Thus, variations that are provided here are standardized. To circumvent this issue and provide an estimate of the AMOC variations in Sverdrup (1 Sv = $10^6$ m³/s), we performed a calibration of the reconstruction from ref. 14. For this purpose, we use the temperature reconstruction over the Holocene[45] over 30–90°N and use the semi-empirical statistical model presented in the next section. We search for the AMOC amplitude that provides the best fit with the temperature reconstruction in terms of Root Mean Square Error (RMSE) and correlation (Fig. 4) using the AMOC fingerprints quantified with five hosing model simulations (see below). The AMOC reconstruction has been extended over the entire Holocene by considering a stable AMOC over the period 0–2 ka, in agreement with a recent study[42]. The same choice has been followed for 12–10 ka, in agreement with ref. 14 for this period.

We then optimize the standard deviation of AMOC variations to best fit with the 30–90°N annual mean reconstruction[45] using the mean of the five statistical methods and 2500 members proposed by this ref. 45. We find a standard deviation of 1.2 Sv as the best calibration, providing a correlation of 0.86 ($P < 0.01$) and a RMSE of 0.23 °C using the LOVECLIM simulation, which is better than for the raw simulations ($r = 0.61$; RMSE = 0.25 °C). When using the TraCE simulation, the correlation between the simulation and the reconstruction increases from 0.38 to 0.63 when adjusted with the AMOC reconstruction, while the RMSE remains similar (Fig. 4).

*Freshwater hosing experiments*. were conducted using five state-of-the-art OAGCMs[44] namely HadCM3, IPSL-CM5A-LR, MPI-ESM, EC-Earth and BCM2. We consider here control simulations, corresponding to historical simulations without any additional freshwater input, and hosing simulations, corresponding to historical simulations with an additional freshwater input of 0.1 Sv released on all the coastal grid points around Greenland with a homogenous rate during 40 years (over the historical era 1965–2004, except for HadCM3 and MPI-ESM for which the experiments were performed over the periods 1960–1999 and 1880–1919, respectively). Results show a weakening of the AMOC at 26°N of 2.6 ± 1.7 Sv associated with the freshwater release, which diminishes the convective activity in the North Atlantic that normally feeds the AMOC in deep water. These simulations are thus capturing the main AMOC fingerprints. We notice that precipitation changes over the Alps associated with a weakening of the AMOC are not significant, despite evidence of enhanced precipitation documented from proxy records. However, regional impacts imply a potential dynamical response that is not adequately resolved by the GCMs. Hence, it is difficult to estimate the exact precipitation response to changes in the AMOC over regions of complex terrain, such as the Alps.

**Semi-empirical model**. To account for reconstructed AMOC variations in the transient simulation, we develop the following linear model:

$$Y = X_{mod} + ßX_{AMOC} \tag{2}$$

where $X_{mod}$ is the temperature from the transient simulation at a given location, $X_{AMOC}$ are the variations of the AMOC multiplied by the fingerprint from the hosing experiment in terms of temperature (or precipitation) and $Y$ is the modeled field, including reconstructed variations in the AMOC. ß is computed from the Holocene reconstructed AMOC variations, adjusted by the amplitude of the AMOC fingerprints from the hosing experiments, where the AMOC decreases on average by 2.6 Sv[44].

## Data availability

The ¹⁰Be data will be deposited at an open-access repository on the informal cosmogenic-nuclide exposure-age database (ICE-D: Alpine; http://alpine.ice-d.org). Semi-empirical AMOC data will be available on request and deposited at the open-access repository NOAA platform.

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

## Acknowledgements

Our colleague and friend Didier L. Bourlès passed away recently. A specialist in cosmogenic dating, he developed ASTER. This article is dedicated to him. Co-authors from IGE acknowledge the support of LabEx OSUG@2020 (Investissements d'avenir—ANR10 LABX56), the International Joint Laboratory GREAT-ICE and the French *Service National d'Observation* GLACIOCLIM (https://glacioclim.osug.fr/) and ANR-15-JCLI-0003-03 BELMONT FORUM PACMEDY. The ASTER AMS National facility (CEREGE, Aix en Provence) is supported by the INSU/CNRS, the ANR through the "Projets thématiques d'excellence" program for the "Equipements d'excellence" ASTER-CEREGE action and IRD. Special thanks to Laetitia Leanni for her help in sample preparation.

## Author contributions

V.J. designed the paper. V.J., A.R., L.C.P.M., P.H.B., T.C., and M.L. realized the field sampling. B.G., J.S., A.G., R.B., L.S., M.L., M.C., G.A., B.D.L., K.K., and V.J. processed the 10Be and 14C sample preparation and analysis. V.J., P.H.B., L.C.P.M., M.L., I.S., I.R., J.S., and B.G. computed the cosmogenic nuclides ages. D.S. ran the AMOC simulations. V.J., D.S., L.M., V.F., and D.V. analyzed forcings. V.J. and V.F. computed Tanar and other patterns. M.V. and Z.H. analyzed Zongo data. V.J. wrote the first draft of the paper and all authors contributed to the discussion and final version of the manuscript.

## Competing interests

The authors declare no competing interests.
