## [Peer Review File · Nature Communications]

In-phase millennial-scale glacier changes in the tropics and north Atlantic regions during the HoloceneReviewers' Comments:

Reviewer #1:

Remarks to the Author:

NOTE: formatted PDF version of this text included as an attached document.

Summary statement:

Overall this paper provides a new model using a Holocene AMOC reconstruction to explain regional glacier fluctuation observed in the paleoclimate record. The chronology and mapping data from their study regions and select other sites reveal a LIA advance that was nearly as large as the early Holocene advances in those areas (with no evidence of a mid-Holocene advance) and further, that these advances correlate in time with a reduced AMOC. Perhaps most compelling is an articulation of our current lack of detailed understanding (both conceptually and numerically) of AMOC variations, which may be an important source of uncertainty in modeling future climate scenarios.

As my expertise is in glacial geomorphology and cosmogenic radionuclide chronologies, not in climate modeling, I focused my attention on making suggestions to improve elements of cosmogenic methods and interpretations communicated here. I also articulated where I did not follow communication about the modeling methods and interpretation that I think might be useful as the authors continue to improve this paper to make it understandable to a wide audience. Certainly, these methods and ideas are of broad interest to many in the geosciences, climatological sciences, and beyond.

Currently, this manuscript reads like a draft rather than a complete final version of a manuscript. I had to work very hard to follow the communication, in comprehending both the text and the figures. Figure labels are missing in many places and figure captions are not complete or as useful as they might be. Further, there is confusing grammar within specific sentences as well as organizationally at the scale of paragraphs (see comments within). I have made specific comments to the text and figures within the pdf document using the comment and edit features. I summarize some of my general comments below.

These new data, modeling efforts, and interpretations stemming from them certainly deserve to be published, however the communication of these elements is not currently written clearly enough to warrant publication at this time. I recommend potential for future publication pending major revision to the text and figures.

The ideal review should answer the following questions:

- What are the major claims of the paper?
 - o AMOC intensity plays a role in glacier mass balance and can account for some of the documented asynchronous glacial advances observed in different global chronologies. These variations in AMOC intensity have potential implications for modeled future climate scenarios.
- Are the claims novel? If not, please identify the major papers that compromise novelty
 - o The authors develop a semi-empirical model that incorporates the related temperature and precipitation changes associated with Holocene AMOC variations in order to explain regional glacier dynamics. They make a strong case that AMOC variations must be considered for the interpretation of the paleoclimate record as well as in future climate forecasting, highlighting a need for the development of more sophisticated transient models.
- Will the paper be of interest to others in the field?
 - o Yes, I believe the paper will be of interest to many in different fields: climatology, paleoclimatology, geomorphology, global change, Quaternary geology, marine science, and any fields considering future climate scenarios.
- Will the paper influence thinking in the field?
 - o For researchers already working on Holocene climate change and specifically the interpretation of paleoglacial environments, the general idea that AMOC variations can influence regional-global climate is not new, however, this paper provides inspiration to work to develop transient simulations with

higher resolution climate models at this moment in time. Beyond enhancing models, we need to interrogate more closely at the growing dataset of glacial chronologies for subtle regional differences that may provide clues to missing or inaccurate components of our climate models. While this type of work is and has been underway (comparing regional/global glacial chronologies) for decades, the growing abundance of high resolution dataset based on exposure ages from glacial features highlights an opportunity as our cosmogenic methods have improved and become a common and accessible tool to many researchers and simultaneously our computing power and climate model sophistication has made it possible to incorporate observations of the somewhat subtle regional differences in dynamic ways that were not possible in the past. This paper is an example of this opportunity being realized.

- Are the claims convincing? If not, what further evidence is needed?
 - o A more thorough review of global glacial datasets would make this paper much stronger – it seems many from the Southern Hemisphere tropics have not been considered (addition comments on this below). In general, I don't find the figures as useful as they might be. The "patterns" that are meant to be identified on Figure 1 would be much clearer and more compelling with a larger review of data (include data from western North America and Asia for example).
- Are there other experiments that would strengthen the paper further? How much would they improve it, and how difficult are they likely to be?
 - o I do not think additional experiments are necessary (although I have limited experience with developing climate models), however I do think consideration of different chronologies in the tropics and beyond would be beneficial (more on this below).
- Are the claims appropriately discussed in the context of previous literature?
 - o Given the length restrictions for this format paper, yes, I believe the claims are appropriately discussed in the context of previous literature. However, additional literature should be incorporated as discussed in other parts of this review.
- If the manuscript is unacceptable in its present form, does the study seem sufficiently promising that the authors should be encouraged to consider a resubmission in the future?
 - o Yes, consider resubmission following major revision (specifically regarding clarification of the text and figures).

For manuscripts that may merit further consideration, it is also helpful if referees can advise on the following points:

- Is the manuscript clearly written? If not, how could it be made more accessible?
 - o I did not find the text of the manuscript to be clearly written. I had to spend a lot of time with the figures especially as they seem to not be labeled clearly and the captions didn't always go far enough to explain the important components of the figure. I found the first ~200 lines of the paper to be most confusing, while I found the Discussion to be more clearly written. I suggest that the introduction and background information to be re-written to provide the necessary context. Some of the background information about AMOC provided in the Discussion might be more useful before the Results. I explain my thoughts on this in more detail below and in comments within the edited document.
- Could the manuscript be shortened to aid communication of the most important findings?
 - o I do not think the main text can be shortened at all. As it is the methods section is very long but all of it is needed (perhaps some can move to supplemental text?) I found the text in lines 448-460 somewhat redundant or not adding much new information. Some of this text could be combined with or replace text in the main text or in figure captions.
- Have the authors done themselves justice without overselling their claims?
 - o Yes, I believe so.
- Have they been fair in their treatment of previous literature?
 - o The authors should include additional cosmogenic chronologies from the tropics as well as areas outside of TANAR (further comments below and within edited doc).
- Have they provided sufficient methodological detail that the experiments could be reproduced?
 - o I cannot comment on this regarding the climate modeling experiments as it is beyond my expertise.
- Is the statistical analysis of the data sound?
 - o I cannot comment on this regarding the climate modeling. The requisite information is provided for the cosmogenic chronologies although I think the authors could comment on why they decided to use

the mean age rather than another option (maximum?) for a moraine age. However, this is not going to make a change to their interpretation as the mean and max ages aren't very different in most cases. It is just common practice.

- Should the authors be asked to provide further data or methodological information to help others replicate their work?
 - o They should make sure to include all information needed to easily recalculate all cosmogenic exposure ages. As our age calculation models will continue to evolve, it is important to always be able to recalculate datasets to keep them relevant. I believe all of the most important information is provided but it is not clearly organized in the tables at present (additional comments within document).
- Are there any special ethical concerns arising from the use of animals or human subjects?
 - o None

General Comments:

A very brief introduction to AMOC would be helpful: general connections to other components of the Earth System, what is known about the paleoAMOC record, and any recent advances in our understanding of AMOC (or at least references provided). As a geomorphologist who studies paleoclimate records and glacial landscapes, I'm very interested in this topic and these new data/interpretations, but would be more engaged and understand more deeply with a bit more of an introduction. Specifically, I found the first 2 paragraphs of the Discussion quite helpful with providing the necessary context for understanding this paper – I think some of this information should be moved before the Results section if possible.

Throughout the text sentences refer to "this" without a clear indication of what "this" is referring to. The authors also use "in line with" often. In some cases, it is very hard to determine what they are using "in line with" to mean exactly. See specific comments within document.

Many sentences early in the paper (lines ~51-63) are dedicated to articulating how some glacial chronologies show a large LIA advance while others do not and that most, world-wide, have little or no record of mid-Holocene advances – these can be shortened to one or two concise and clear sentences supported by clarifying Figure 1.

Comments about specific locations within the text:

Lines 51-53: Confusing statement:

"...consistent with this summer insolation forcing, it is generally assumed that northern hemisphere glaciers increased in extent throughout the mid and late Holocene, recording their maximum size during the Little Ice Age (LIA), while most southern hemisphere glaciers gradually retreated until a minimum ice extent was reached during the late Holocene" (direct text from manuscript)

Four references were cited for this statement of "generally assumed" information; these references are not review studies that span many regions on Earth but rather studies that are spatially and temporally limited and therefore are not really representative of a broad view supporting a general assumption. While I think there are MANY studies that do support the notion the authors describe, the selection of references is limited to studies in only a few areas. At the very least, ONE study cited here should probably involve the tropics!

- Schaefer et al., 2009: New Zealand and European/Alaskan glaciers during last 4ka
- Putnam et al., 2012: New Zealand and European glaciers during the Holocene
- Kaplan et al., 2013: Europe and New Zealand glaciers during the Holocene
- Reynhout et al., 2019: Patagonia and New Zealand glaciers during the Holocene

Lines 56-57: The authors then go on to say that this general assumption doesn't really hold up when you look at a "handful" of studies (although they cite the same number of references spanning a similar number of areas) and go on to say the tropics are "enigmatic" and that no mid-Holocene moraines are known. There are a few mid-Holocene moraines and glacial landforms (bogs behind moraines), but it's true that there aren't many. Some cosmogenic ages on mid-Holocene moraines

exist in the Cordillera Blanca of northern Peru – the region on Earth with the largest volume of tropical glaciers but seems to be notably absent from discussion in this manuscript. However, at this point most of these data are only published in the abstract literature. The review paper Rodbell et al., (2009) logs an extensive 14C (buried peat) and 10Be (moraine boulders) record of Holocene glaciers in the tropics with few mid-Holocene records (although these are mainly from 14C in bogs/lakes rather than CRE on moraines). Stansell et al. (2017) notes one mid-Holocene CRE on a moraine in the Cordillera Blanca. There are in fact data points from this region on Fig 1 but it isn't clear from the text, figure caption, or cited references that the authors rigorously explored the data available from the Southern Hemisphere tropics. The easiest thing to do is to say instead of "no mid-Holocene moraines are known" say "few mid-Holocene moraines have been identified and directly dated".

Overall, this whole section would be improved with these modifications:

- 1) Clearly articulate that they are only talking about alpine glaciers and show all of the records referred to on the map on Fig 1. Clarify that the tropical glaciers and mid-latitude glaciers of the same hemisphere are not necessary in phase.
- 2) The statement about it being generally accepted that LIA ice was the biggest of the Holocene in NH glaciers should be supported by references from throughout the NH, not just European glaciers (as it is the "handful" outside of Europe is supporting just the opposite!)
- 3) Clarify that the main point is that the Holocene ice fluctuation was generally out of phase at different latitudes: perhaps separate the tropical glaciers from the mid-latitude glaciers in this discussion in a clearer way. Right now, the blanket statement about what is generally accepted is based on only mid-latitude glaciers in New Zealand, Patagonia, and Europe.
- 4) Handle the tropics in a more thoughtful way – at least articulate that there are locations where the LIA might be most extensive advance of Holocene while in other locations there is no evidence of a LIA advance and still others where there is a LIA advance but it is less extensive than the early Holocene advance (in some cases the LIA is MUCH less extensive in other cases it is nearly as extensive as the early Holocene advance). Rodbell, Smith, and Mark (2009) and other compilations such as this might be a good way to reference more extensive datasets with fewer direct references. Rodbell et al. (2009): "The spatial-temporal pattern of Holocene glaciation features tantalizing but incomplete evidence for an Early to Mid- Holocene ice advance (s) in many regions except the arid Andes surrounding the Bolivian Altiplano. In this latter region, the LIA or a slightly older advance that occurred within the last millennium was the most extensive advance of the Holocene. In many regions, there is strong evidence for Neoglacial advances in the interval between 1.0 and 2.5 ka. Moraines dating to the LIA are seen in all presently glacierized mountain ranges; most of these date to within the past 0.45 ka. In many regions, a more extensive advance occurred several hundred years prior to the onset of the LIA."

Figure 1 is hard to understand without a thorough read of the methods and supplemental data. For example, what does a NH or SH "pattern" refer to? The white circles are called "NH pattern" although the caption does not say what this means – what is the "pattern"? Some of the locations with white circles are in the Southern Hemisphere. Maybe they are trying to say that those glaciers are behaving like others in the NH? Is this "pattern" based on only the new data presented here or on a review of the literature available? There needs to be a clear articulation of what these "patterns" are – perhaps in the caption or in the main text (Results?). Overall, I think this general idea is very compelling. I would like to see a thorough review of all of the studies available with a comprehensive figure showing the same type of info the authors are trying to show, but not just based on 1-2 chronologies in each spot. Even if a generalize figure is included in the main text, a comprehensive and easier to read (larger!) figure should be in the supplemental material as it is one of the foundational observations leading to their hypothesis.

Line 140: At this point in the paper, there isn't a clear picture of what the TANAR glacier evolution is. As the authors articulate in lines 119-121, there are some similarities among the glaciers studied here, but it hasn't been established by the authors that this is a pattern shared by most or ALL TANAR glaciers – in fact it seems most of the tropical glaciers DO NOT fit this pattern (or only partially fit it)

in that the late Holocene advance wasn't close to as extensive as the late Holocene advance. However, this observation, that in some valleys the ~9-10ka moraines are just outboard of the LIA moraines is really very interesting and important. But equally important is documenting exactly WHERE this is the case and where it is not. This cannot be done using only evidence from just 1-2 locations in a region (Bolivia) to represent the whole tropical region (e.g. Fig 2b). Although maybe beyond the extent of this study, something that might help to articulate a "pattern" might be to consider valley morphology (slope, hypsometry, contributing area) when viewing the distribution of the normalized distance between LIA and early Holocene moraines.

Lines ~148-154 and 455-460: It seems important to note that the paleo-oscillations (PD and ENSO) are correlated with positive paleo-glacier mass balance opposite to the modern relationship between ENSO/PD and glacier mass balance. However, the records discussed here are both pretty short records. Perhaps some longer ENSO records exist in some western Andean studies that might be useful (archeological, debris flows, marine sedimentation)? I wonder how reliable comparison to the modern is given post-industrial forcings. How do other proxy records for SST, paleo-precipitation, ENSO/PD compare to each other? I mean, is the post-industrial period the outlier? Also, the text in lines 148-154 seems to say more or less the same thing as 455-460. Perhaps one can be removed or they can be combined in to one more concise discussion in either the main text or Methods.

Lines 327+: $^{14}\text{C}/^{10}\text{Be}$ Numerical Model – I would like to know more about this – at least show a figure with the conceptual view of the burial (and erosion) and exposure history of the bedrock samples in the supplemental data. The plot of erosion probabilities isn't extremely useful without more context and explanation.

Line 430-431: In this section the authors talk about how they compare their new chronologies to existing chronologies. They chose to compare their data from the tropics to only two areas outside of the tropics (New Zealand and Patagonia). This seems quite limiting. Why not at least compare their data to one other tropical region (namely, the location with the largest volume of tropical glaciers and is located just to the north of their study site)? It seems they also chose to only compare their data where there were locations that had chronologies similar to theirs. They say: "We only selected glacial valleys where the front position was well constrained for the early and late Holocene periods, without identified undated moraines in-between." It seems that for a paper making global-scale interpretations and considering climate phenomena affecting both hemispheres, they should be broader in their review of existing chronologies – both for the background information (as described above) and for comparing their new chronologies to other regions.

Cosmogenic chronology: a few things to improve

- 1) Tables are very hard to read at present – align column headings with data, clarify units, and include ALL INFORMATION needed to recalculate the ages (most info seems to be there such as shielding, elevation, etc., however organizing it a bit more clearly would be helpful.)
- 2) Be a bit more descriptive about each geomorphic feature sampled and describe any assumptions considering the sample location. For example – in sampling the roches moutonnées: a) where were these landforms located with respect to the glacial system?: below the ice in the valley bottom? On a ridge? Would they be the last to be ice-free? b) what is the expectation for what the cosmogenic radionuclide concentrations (ages) measured from these landforms mean?: the time since ice no longer buried the bedrock in that location? a combination of exposure and erosion? Or does the concentration only reflect erosion by ice? The authors do discuss this a bit and certainly the reason why they are using two isotopes is to get at the complicated burial history, but their modeled scenario isn't clearly articulated (a figure would help – S7 isn't useful without more context). Further, how was muogenic production dealt with? It seems maybe it wasn't. How does changing ice thickness affect the production rate in the rock below? Was this considered? If not, explain why.
- 3) The sample names should always be presented in the same format in the paper, figures, and tables. Glacial landform terminology should also be consistent: decide on terminus, front, toe, etc.

Reviewer #2:
Remarks to the Author:

Review of "AMOC controls on millennial-scale glacier changes during the Holocene"

The study challenges the assumption of increased/reduced NH/SH ice sheet extent from early-Holocene to Little Ice Age (LIA) as caused by the corresponding insolation changes, by introducing new CRE which show a retreat of ice sheets from early-Holocene to mid-Holocene, followed by a re-advance in the late-Holocene, for both tropical Andes and North Atlantic regions. Using reconstruction data and model simulations, the authors report that such evolution of ice sheet extent is associated with millennial-scale temperature and precipitation variances linked to AMOC changes.

The author propose that the AMOC strength, rather than the summer insolation is the primarily forcing factor of the glacier during the Holocene. However, I think the evidence they have provided is not robust to support their conclusions. In particular, I have two major concern on their results.

1. I agree with the author that glacier primarily responses to a temperature change in summer. However, AMOC variability primarily manifest temperature anomaly in winter. The author should compare the summer temperature anomaly related to the AMOC variability with that induced by insolation changes. I believe that insolation has more pronounced effect than the AMOC does.

2. The glacier has long response time to temperature change, the equilibrium retreat of glacier in response to warmer climate can be thousands of years. Therefore, the minimum extend of glacier will of course take place later than the temperature peak (e.g., Briner et al., 2016). The shape difference of insolation and glacier extend can also be induced by such delay response.

In other words, the advance/retreat of ice sheet extent does not directly depend on the cooling/warming of the climate, but on the background temperature which is below/above the threshold to cause the advance/retreat of ice sheet extent. In my opinion, though the NH summer insolation was reducing from early to mid Holocene, the temperature might still stay in a range leading to retreat of glacier. The threshold was reached at around mid to late Holocene, from then the ice sheet begin to re-advance. Therefore there is a delaying effect of the peak-insolation or temperature on the occurrence of minimum ice sheet extent. The author should at least consider this delaying effect when examining the affecting factors of ice sheet changes.

reference

Briner, Jason P., et al. "Holocene climate change in Arctic Canada and Greenland." *Quaternary Science Reviews* 147 (2016): 340-364.

Response to reviewers' comments

For your convenience, our response in Times bold. Blue highlighted sentences correspond to text modified in the manuscript.

Reviewer 1

Before answering the remarks by R1, we present the criteria that were used to select glacier records in this study. We used the global Ice-D (<http://alpine.ice-d.org/>; <https://doi.org/10.5194/gchron-2020-6>) cosmogenic database developed by G. Balco and looked for any glacier that would fulfill these criteria. In the revised version, we clarify these criteria. In addition, as suggested by R1, we considered a global overview of glacier records.

In this paper, we explored forcings controlling glacier fluctuations at a millennium time scale during the Holocene. Summer insolation is suggested to have a significant influence over the whole Holocene. It reached a minimum value during the last centuries in the northern hemisphere. As a glacier obliterates previous moraines formed during less extensive advances, moraines dated to the early or mid-Holocene cannot be observed in a catchment if the glacier is mainly driven by this forcing. Consequently, the LIA moraines must correspond to the maximum Holocene extent. In order to test this hypothesis, we focused on mountain glaciers, ignored ice sheet outlets and selected relevant chronologies based on 4 criteria.

- 1- We did not consider catchments with moraines that may correspond to the early Holocene or an older period such as the Younger Dryas if uncertainties associated with the chronologies are considered.
- 2- We did not select catchments with undated moraines in-between the LIA and late glacial moraines. If another superimposed forcing factor played a significant role, pre-LIA moraines may have formed. Following this strategy, catchments with obvious undated moraines immediately downslope from the LIA front position were ignored (as these undated moraines may have been formed either during the mid-Holocene or during the early Holocene and thus may reflect the influence of other forcings).
- 3- We did not take into account landforms with only one CRE age or unpublished data.
- 4- Poorly constrained moraines with a large dispersion in the dataset were not considered.

The conclusion of this global review revealed three distinct patterns

- 1- Glaciers that were investigated in the previous version of the draft reflect the influence of AMOC forcing superimposed on insolation changes in TANAR (glaciers from the Tropical Andes and Northern Atlantic Regions).
- 2- There are few glacial records, mostly located in Alaska, that show a Holocene maximum extent during the pre-LIA Neoglacial. These chronologies do not reflect the role of insolation only. We assume this Neoglacial advance may correspond to the influence of AMOC changes (Fig. 4), however, the lack of early Holocene advances remains enigmatic.
- 3- Finally, Asian glaciers and southern hemisphere mid-latitude glaciers exhibit other patterns of change than TANAR, particularly when including moraines from the mid-Holocene

General Comments:

A very brief introduction to AMOC would be helpful: general connections to other components of the Earth System, what is known about the paleoAMOC record, and any recent advances in our

understanding of AMOC (or at least references provided). As a geomorphologist who studies paleoclimate records and glacial landscapes, I'm very interested in this topic and these new data/interpretations, but would be more engaged and understand more deeply with a bit more of an introduction. Specifically, I found the first 2 paragraphs of the Discussion quite helpful with providing the necessary context for understanding this paper – I think some of this information should be moved before the Results section if possible.

We certainly appreciate this comment, but we had a different approach in mind that we would like to preserve. The idea of the paper is to verify the dominance of insolation as the key driver of long-term glacier fluctuations during the Holocene. To do so, our approach entails testing different possible forcings without any prior assumptions. Initially, there is no reason to assume that AMOC is the key driver of tropical glacier fluctuations. ENSO or PDO for instance could be expected to provide a more suitable forcing without a priori knowledge of the conclusions our paper provides. Therefore, we prefer not to start out with a long paragraph about the AMOC, as it is inconsistent with our narrative and premise and may mislead the reader. However, we added some text about potential climate impacts of the AMOC early in the introduction, in order to justify why we considered this forcing, and thus indirectly “provide the necessary context for understanding this” forcing as recommended by R1.

Throughout the text sentences refer to “this” without a clear indication of what “this” is referring to. The authors also use “in line with” often. In some cases, it is very hard to determine what they are using “in line with” to mean exactly. See specific comments within document.

We agree and we have improved this aspect.

Many sentences early in the paper (lines ~51-63) are dedicated to articulating how some glacial chronologies show a large LIA advance while others do not and that most, world-wide, have little or no record of mid-Holocene advances – these can be shortened to one or two concise and clear sentences supported by clarifying Figure 1. Comments about specific locations within the text.

We believe it is important to present the context about the influence of insolation on glacier fluctuations because this is the central question of this paper.

Lines 51-53: Confusing statement:“...consistent with this summer insolation forcing, it is generally assumed that northern hemisphere glaciers increased in extent throughout the mid and late Holocene, recording their maximum size during the Little Ice Age (LIA), while most southern hemisphere glaciers gradually retreated until a minimum ice extent was reached during the late Holocene” (direct text from manuscript); Four references were cited for this statement of “generally assumed” information; these references are not review studies that span many regions on Earth but rather studies that are spatially and temporally limited and therefore are not really representative of a broad view supporting a general assumption. While I think there are MANY studies that do support the notion the authors describe, the selection of references is limited to studies in only a few areas. At the very least, ONE study cited here should probably involve the tropics!

- Schaefer et al., 2009: New Zealand and European/Alaskan glaciers during last 4ka
- Putnam et al., 2012: New Zealand and European glaciers during the Holocene
- Kaplan et al., 2013: Europe and New Zealand glaciers during the Holocene
- Reynhout et al., 2019: Patagonia and New Zealand glaciers during the Holocene

We agree with R1. Actually, we avoided review papers in order not to be criticized by a potential reviewer who would prefer original studies. We have now changed this and include both the global review (Solomina et al., 2015) and the regional review of the tropical glaciers (Rodbell et al., 2009). Moreover, we claim that our analysis is based on the most complete database of CRE ages and includes data from all over the world. Finally Fig 1 in the submitted version was already based on Schaefer et al., (2009); Putnam et al., (2012) and Reynhout et al., (2019).

Lines 56-57: The authors then go on to say that this general assumption doesn't really hold up when you look at a “handful” of studies (although they cite the same number of references spanning a similar number of areas) and go on to say the tropics are “enigmatic” and that no mid-Holocene moraines are known. There are a few mid-Holocene moraines and glacial landforms (bogs behind moraines), but it's

true that there aren't many. Some cosmogenic ages on mid-Holocene moraines exist in the Cordillera Blanca of northern Peru – the region on Earth with the largest volume of tropical glaciers but seems to be notably absent from discussion in this manuscript. However, at this point most of these data are only published in the abstract literature. The review paper Rodbell et al., (2009) logs an extensive ^{14}C (buried peat) and ^{10}Be (moraine boulders) record of Holocene glaciers in the tropics with few mid-Holocene records (although these are mainly from ^{14}C in bogs/lakes rather than CRE on moraines). Stansell et al. (2017) notes one mid-Holocene CRE on a moraine in the Cordillera Blanca. There are in fact data points from this region on Fig 1 but it isn't clear from the text, figure caption, or cited references that the authors rigorously explored the data available from the Southern Hemisphere tropics. The easiest thing to do is to say instead of “no mid-Holocene moraines are known” say “few mid-Holocene moraines have been identified and directly dated”.

Our analysis is based on the global online Ice-D database [http://alpine.ice-d.org/allsites/South America](http://alpine.ice-d.org/allsites/South_America) and we searched for mid-Holocene moraine ages in the tropical Andes, including Peru and Bolivia. We did not find any moraine dated to the mid-Holocene. This analysis agrees with our previous regional review (see Jomelli et al., (2014 Fig. 2 and Supplementary Tables S1-4) with an update of all ages published before 2014). There are two exceptions that concern 2 samples only. We found one sample from Peru published by Licciardi et al., (2009) (sample PE08-13 dated to 5.3 ± 0.3) which was already cited in this paper. The second exception comes from Stansell et al. (2017) which is also cited in this paper. The authors report a moraine dated by four samples. One sample yielded an age of 6.2 ± 0.3 ka while the 3 other samples collected on the same moraine are dated to the late Holocene. The two samples from the mid-Holocene, one from Licciardi et al. (2009) and the other from Stansell et al. (2017), are thus exceptions in the database and may be considered outliers. We assume this difference of opinion with R1 may be related to the new cosmogenic nuclide production rates, scaling models, and radiocarbon calibration published in recent years that affected previous chronologies reported by Rodbell et al. (2009). However, we agree with R1 regarding ^{14}C from peat bogs that suggest possible mid-Holocene advances, but ages reported in Rodbell et al. (2009) show only maximum or minimum ages, which are not relevant for our purpose.

Overall, this whole section would be improved with these modifications: 1) Clearly articulate that they are only talking about alpine glaciers and show all of the records referred to on the map on Fig 1. Clarify that the tropical glaciers and mid-latitude glaciers of the same hemisphere are not necessary in phase. **We agree. We already mentioned in the original manuscript that we focus exclusively on mountain glaciers. We have now clarified this further (see line 47 in the revised version): ‘Small (< 100 km²) mountain glaciers are particularly relevant for such an analysis and are the sole target of this study. We report new *in situ* ^{14}C and ^{10}Be cosmic ray exposure (CRE) ages from alpine glacial valleys in the tropical Andes (Bolivia), the French Alps, Greenland and the Cascade Range (USA), along with other published glacier records (methods), to document alpine glacier evolution during the Holocene at a global scale. Based on CRE moraine ages we show that in the tropical Andes and the Northern Atlantic Regions (TANAR) glaciers experienced a similar behavior, which differs from the other regions of both hemispheres.’**

3) Clarify that the main point is that the Holocene ice fluctuation was generally out of phase at different latitudes: perhaps separate the tropical glaciers from the mid-latitude glaciers in this discussion in a clearer way. Right now, the blanket statement about what is generally accepted is based on only mid-latitude glaciers in New Zealand, Patagonia, and Europe.

We agree and have modified the text accordingly. A global review was performed and is now discussed in the paper, including glaciers from Asia and the US.

4) Handle the tropics in a more thoughtful way – at least articulate that there are locations where the LIA might be most extensive advance of Holocene while in other locations there is no evidence of a LIA advance and still others where there is a LIA advance but it is less extensive than the early Holocene

advance (in some cases the LIA is MUCH less extensive in other cases it is nearly as extensive as the early Holocene advance).

We do not fully agree with this statement. Based on CRE ages, there are no glaciers which show a larger LIA advance than during the early Holocene. However, we fully agree with R1 about the amplitude of glacier extents. Indeed, the distance between the early Holocene and the LIA moraines depends on the catchment characteristics, its topographic and geomorphological context. This difference is considered in Fig. 1, albeit smoothed, as we calculated a regionally averaged value.

Rodbell, Smith, and Mark (2009) and other compilations such as this might be a good way to reference more extensive datasets with fewer direct references. Rodbell et al. (2009): “The spatial–temporal pattern of Holocene glaciation features tantalizing but incomplete evidence for an Early to Mid- Holocene ice advance (s) in many regions except the arid Andes surrounding the Bolivian Altiplano. In this latter region, the LIA or a slightly older advance that occurred within the last millennium was the most extensive advance of the Holocene. In many regions, there is strong evidence for Neoglacial advances in the interval between 1.0 and 2.5 ka. Moraines dating to the LIA are seen in all presently glacierized mountain ranges; most of these date to within the past 0.45 ka. In many regions, a more extensive advance occurred several hundred years prior to the onset of the LIA.”

We agree with Rodbell et al. (2009) but our approach is based on direct dating only, from CRE ages on moraine records. As the position of the glacier between two moraines is unknown as long as in situ ¹⁴C ages do not permit constraining the position of the terminus, we can assume that minor glacier advances occurred during the mid-Holocene. However, direct dating proves that these possible minor glacier advances were obliterated by late Holocene advances. We also fully agree with the fact that Neoglacial advances are observed. This is exactly what we documented on one of the investigated glaciers (see Charquini North for instance (Fig. S2)).

Line 140: At this point in the paper, there isn't a clear picture of what the TANAR glacier evolution is. As the authors articulate in lines 119-121, there are some similarities among the glaciers studied here, but it hasn't been established by the authors that this is a pattern shared by most or ALL TANAR glaciers – in fact it seems most of the tropical glaciers DO NOT fit this pattern (or only partially fit it) in that the late Holocene advance wasn't close to as extensive as the late Holocene advance. However, this observation, that in some valleys the ~9-10ka moraines are just outboard of the LIA moraines is really very interesting and important. But equally important is documenting exactly WHERE this is the case and where it is not. This cannot be done using only evidence from just 1-2 locations in a region (Bolivia) to represent the whole tropical region (e.g. Fig 2b). Although maybe beyond the extent of this study, something that might help to articulate a “pattern” might be to consider valley morphology (slope, hypsometry, contributing area) when viewing the distribution of the normalized distance between LIA and early Holocene moraines.

We believe there is a misunderstanding, probably related to the fact that we did not sufficiently explain what we consider as the TANAR pattern. The TANAR pattern refers to a glacial trend during the Holocene that is consistent with AMOC changes. It only concerns the age of the largest glacier advance during the Holocene and not the distance of the moraines during the Late Holocene. When the largest advance of a glacier occurred in the early Holocene and the second largest during the Holocene including the LIA, we assume it corresponds to the TANAR pattern. The distance between moraines from the early to the late Holocene or during the late Holocene is not considered. We agree with R1 about the influence of geomorphic factors on the distance between moraines and do not consider this point as discriminant in our list of criteria given at the beginning of this response. Based on these assumptions we claim that the existing data in the global online Ice-D database http://alpine.ice-d.org/allsites/South_America confirms that all documented glaciers (n=10) in Bolivia and Peru do show a TANAR pattern. As their evolution shows a good correspondence with AMOC changes, we replaced TANAR pattern in Fig. 1 with AMOC pattern.

Lines ~148-154 and 455-460: It seems important to note that the paleo-oscillations (PD and ENSO) are correlated with positive paleo-glacier mass balance opposite to the modern relationship between ENSO/PD and glacier mass balance. However, the records discussed here are both pretty short records. Perhaps some longer ENSO records exist in some western Andean studies that might be useful (archeological, debris flows, marine sedimentation)?

It is important note that ENSO or the PDO per se do not induce long-term mass balance changes in tropical Andean glaciers. This only occurs if these modes persist in one of their extreme phases (El Niño or La Niña) for a longer period of time. Hence for Pacific SST to induce glacier advances would require either a shift in the mean state or increased variability that is skewed toward either the positive or negative phase of ENSO or the PDO. Unfortunately, there is no agreement in the literature on how ENSO or the PDO changed during the course of the Holocene, as these studies originate from different regions in the Pacific domain, rely on different proxies, cover different time periods, are characterized by different temporal resolution and climate sensitivity or dependent on different teleconnection mechanisms that may or may not have remained stationary over time. We have cited some of the seminal papers on this aspect in our original submission. While there certainly are additional studies that could be cited, they do not add anything pertinent to this discussion and in our opinion, would only distract from the main focus of this study, which is the role of the AMOC.

I wonder how reliable comparison to the modern is given post-industrial forcings. How do other proxy records for SST, paleo-precipitation, ENSO/PD compare to each other? **As discussed above, there is indeed little agreement among proxy-based reconstructions. For that reason, we only focused on longer records that document both the variance of ENSO and the frequency.**

I mean, is the post-industrial period the outlier? Also, the text in lines 148-154 seems to say more or less the same thing as 455-460. Perhaps one can be removed or they can be combined in to one more concise discussion in either the main text or Methods. **In the main text, we focus on ENSO, for which several reconstructions exist, while in the Methods section we briefly discuss questions about a possible PDO influence on glacier changes. While the ENSO influence on glacier mass and energy balance is quite well understood, this is much less the case for the PDO, as existing glacier mass balance records are too short to fully investigate the influence of Pacific multidecadal variability. For that reason, we prefer not to change the text.**

Lines 327+: ¹⁴C/¹⁰Be Numerical Model – I would like to know more about this – at least show a figure with the conceptual view of the burial (and erosion) and exposure history of the bedrock samples in the supplemental data. The plot of erosion probabilities isn't extremely useful without more context and explanation.

We agree. We have added more information and discussion about this aspect in the revised manuscript. We also changed the Fig. S8 as suggested by R1.

Line 430-431: In this section the authors talk about how they compare their new chronologies to existing chronologies. They chose to compare their data from the tropics to only two areas outside of the tropics (New Zealand and Patagonia). This seems quite limiting. Why not at least compare their data to one other tropical region (namely, the location with the largest volume of tropical glaciers and is located just to the north of their study site)? It seems they also chose to only compare their data where there were locations that had chronologies similar to theirs. They say: "We only selected glacial valleys where the front position was well constrained for the early and late Holocene periods, without identified undated moraines in-between." It seems that for a paper making global-scale interpretations and considering climate phenomena affecting both hemispheres, they should be broader in their review of existing chronologies – both for the background information (asdescribed above) and for comparing their new chronologies to other regions.

We agree with the fact that our criteria were not clear enough and changes were made accordingly as detailed at the beginning of the response letter.

Cosmogenic chronology: a few things to improve¹) Tables are very hard to read at present – align column headings with data, clarify units, and include ALL INFORMATION needed to recalculate the ages (most info seems to be there such as shielding, elevation, etc., however organizing it a bit more clearly would be helpful.)

Changes were made accordingly.

2) Be a bit more descriptive about each geomorphic feature sampled and describe any assumptions considering the sample location. For example – in sampling the roches moutonnées: a) where were these landforms located with respect to the glacial system?: below the ice in the valley bottom? On a ridge? Would they be the last to be ice-free? b) what is the expectation for what the cosmogenic radionuclide concentrations (ages) measured from these landforms mean?: the time since ice no longer buried the bedrock in that location? a combination of exposure and erosion? Or does the concentration only reflect erosion by ice? The authors do discuss this a bit and certainly the reason why they are using two isotopes is to get at the complicated burial history, but their modeled scenario isn't clearly articulated (a figure would help – S7 isn't useful without more context). Further, how was muogenic production dealt with? It seems maybe it wasn't. How does changing ice thickness affect the production rate in the rock below? Was this considered? If not, explain why. 3) The sample names should always be presented in the same format in the paper, figures, and tables. Glacial landform terminology should also be consistent: decide on terminus, front, toe, etc.

Changes were made accordingly. See the new Fig. S8. We also changed the text related to the ¹⁴C ages in the method to make it clear.

Figure 1 is hard to understand without a thorough read of the methods and supplemental data. For example, what does a NH or SH “pattern” refer to? The white circles are called “NH pattern” although the caption does not say what this means – what is the “pattern”? Some of the locations with white circles are in the Southern Hemisphere. Maybe they are trying to say that those glaciers are behaving like others in the NH? Is this “pattern” based on only the new data presented here or on a review of the literature available? There needs to be a clear articulation of what these “patterns” are – perhaps in the caption or in the main text (Results?). Overall, I think this general idea is very compelling. I would like to see a thorough review of all of the studies available with a comprehensive figure showing the same type of info the authors are trying to show, but not just based on 1-2 chronologies in each spot. Even if a generalize figure is included in the main text, a comprehensive and easier to read (larger!) figure should be in the supplemental material as it is one of the foundational observations leading to their hypothesis.

Thank you for these remarks. Changes were made accordingly. For instance, we replaced ‘NH Pattern’ with ‘AMOC pattern’, referring to a glacier evolution that shows a good correspondence with AMOC changes. In contrast, some other glaciers are characterized by a different patterns of change, such as glaciers in the Himalayas. Moreover the list of investigated glaciers (n= 66) is given in Table S2.

Comments from the PDF

Abstract comments

Line 31: This is not quite shown here - there needs to be a more extensive review of glaciers OUTSIDE these regions - Asia, Africa, western NA...At least include the in Fig 1.

We agree. We now include other regions as well.

Line 36: This is confusing: the paper mostly articulates now the mid-Holocene has minimal evidence of glacier advances while the larger advances are during the early and late glacial. This sentences seems contradictory to parts of the main text. Should this be strong and retreat or instead of mid-Holocene, early and late?

Yes, we fully agree. This was a mistake, which now has been corrected.

Main text

Line numbers refer to the submitted version. Correspondence with the revised version was given when necessary.

Lines- 51-54. **We agree and use the global database from Ice-D (<http://alpine.ice-d.org/>)**

Line 57: **Done, thanks for your suggestion.**

Line 61: There are a few mid-Holocene in the C Blanca of Peru I believe...see Rodbell et al., 2009.

We have already addressed this issue in our response above. We have limited our analysis to direct moraine dating rather than chronologies based on ^{14}C from peat bogs that only propose minimum or maximum ages which cannot guarantee a robust chronology. The few cases observed in the Cordillera Blanca correspond to this later case. Moreover, aside from the two samples discussed above, we did not identify any CRE chronologies documenting a mid-Holocene moraine.

Line 75-76: **We agree and again thank you for your suggestion.**

Line 95: Where are these? In the valley? I can't tell if the idea in sampling these was to determine if ice was in the valley or covering a high elevation point? Roches moutonnees landforms are not specific to a location in a valley or on a ridge - it is a landform with a specific shape formed by subglacial processes. More clarity is needed here to understand what this age would represent. Was the sample collected to determine when that bedrock was ice-free? Is this bedrock in the valley floor or well above?

We fully agree with these remarks and now include more details about the location (see lines 105-108). *'In situ ^{14}C CRE ages from roches moutonnées located in between the current front position and the late Holocene moraines on Charquini North independently reveal that during the mid-Holocene the front of the glacier was positioned up-valley from its LIA extent'* . Moreover the dedicated section in method was updated and a new Fig. 8 was drawn.

Line 99: **Done. We now write: "after the deglaciation that started about 19 000 years ago"**

Line 101: Does this mean the maximum event EVER or the maximum extent of the Late Holocene - and where is this extent relative to the early Holocene extent?

Here we distinguish the maximum Holocene extent (line 115 in the new draft) from the late Holocene maximum extent, which is, however, smaller than the early Holocene stage. As seen in Fig. S1, S2, the early Holocene moraine is very close to, but located downslope of, the late Holocene moraines. If this were not the case, then early Holocene moraines would have been destroyed by the late Holocene advance and no longer be identifiable in the field.

Line 115: This is vague and passive. Do you mean that this new dataset documents reduced glacier extent during the mid-Holocene similar to records in the Alps (refs) - clarify.

Thank you for your comment. Our intention was to point out that several previous papers demonstrated a clear glacier retreat during the mid-Holocene. We clarified this sentence (see line 128). We now write: *'Moreover, previous studies^{1,22} showed reduced glacier extents during the mid-Holocene in the Alps between 10 ka and 4 ka.'*

Line 119-121. This seems at odds with text in the abstract.

Thank you, we clarified the sentence in the abstract.

Line 143: It would be helpful to articulate this "pattern" within the text of Fig 1 caption and clarify the figure to make this pattern more clearly visible.

Yes, you are right. Thank you for this suggestion. We clarified this pattern in the caption of Fig. 1.

Line 152-153. I don't follow as written. "Not in line with" is not a very useful phrase here. Do you mean the El Niño conditions are not correlated in time with ice advances (because they seem to me to be...late glacial and LIA: both times of advance and both with El Niño-like conditions = "in line") - The El Niño conditions temporally are aligned...but maybe what you mean is that, paradoxically, the periods of paleoglacier advance are correlated in time with El Niño-like conditions and that, at present, El Niño-like conditions are associated with negative mass balance, not positive mass balance.

Indeed, this is exactly what we mean. 'Niño-like' equals a negative mass balance and is therefore inconsistent with a glacier advance. We changed the sentence accordingly and now write: However, recent investigations of both variance and frequency of paleo-ENSO proxies during the Holocene^{25,29} suggest El Niño-like conditions during the early Holocene and LIA, which do not support a direct relationship with major glacier advances documented for these two periods²⁵⁻²⁹. ENSO variability can thus not explain the millennial glacier trend in the Andes.

Line 160: Could you provide a bit of information on ideas to what causes an "enhanced AMOC"?

The causes remain unclear. Ice sheet changes play a significant influence but internal variability is also considered. The exact processes leading to a mid-Holocene AMOC strengthening cannot be constrained in this study and could result from changes in atmospheric forcing, sea-ice cover, hydrological balance for instance (e.g. Drijfhout et al., 2013). Holocene AMOC changes would result from the adjustment of the northern North Atlantic surface buoyancy to the varying boundary conditions.

Line 164. Unclear phrasing: What does "related with" mean here. Cannot follow.

Those directly impacted by AMOC changes. This has been clarified (see line 195).

Line 166: Which, SST and AMOC or glacial mass balance and AMOC or glacial mass balance and SST? Unclear what "this" refers to here. **Done - please see lines 195-197.**

Line 168: Unclear: **Done - please see lines 195-197.**

Line 172: This paragraph is in general really helpful and I think it would be beneficial to move this up before the results - just after line 80 as it clarifies what the hypothesis is. However Figure 3 and S5 and S6 could be more helpful with clear units on the figures and description within caption.

We thank reviewer 1 for this interesting suggestion regarding the position of this paragraph. Please see our response made earlier in this document.

Line 178. perhaps identify this on S5 - could include a line indicating a threshold for AMOC intensity. A figure (cartoon) showing these relationships would be quite helpful but could be done with better labeling and description on existing figures.

We think there is a misunderstanding here. In Fig. 2 the curve in G already shows this. Fig. S5 updated to Fig. S6 in the revised version corresponds to temperature simulated by GCMs related to the corrected AMOC. We clarified this in the text and in the caption, as recommended.

Line 181-183. This is opposite of a statement in the abstract line 36.

Yes we agree. This was a mistake that we have corrected.

Line 188-190_This is not clear on figure 4 - articulate in caption. Add labels and units to make figures understandable.

Thank you. The black curve in Fig. 4c,d,e corresponds to temperature and precipitation from the two GCMs (without AMOC correction), while the blue curve corresponds to temperature and precipitation after the correction of AMOC intensity has been applied. This has been changed in the figure caption.

Line 231: This sentence is hard to follow - too many "from"s Over the whole area? No over the TANAR exclusively as mentioned in the text.

We agree and changed the sentence accordingly. We now write (line 263-264). 'This semi-empirical model was based on the three recent AMOC reconstructions and AMOC fingerprints.

Lines 236-238 Where - show us on a figure by explicitly explaining in the caption and using labels on the figure. I think showing the timing of advance on Fig 4, potentially with gray pars on plots c-e, would be helpful.

Thank you. Please see our comment related to Fig. 4, 7 lines above . This is shown by the difference between the black and blue curves in Fig. 4c,d,e. For instance, in the tropical Andes the increase in precipitation during the Late Holocene related to a corrected AMOC in the model (Fig 4e) corroborates glacier advances and is in agreement with proxy records in the region (e.g. Stansell et al., 2017).

Line 255-256. Include references here.

This is shown in Fig 4. In fact, models without corrected AMOC forcing do not show any change in precipitation during the Holocene which is inconsistent with proxy records (e.g. Stansell et al., 2017) in the tropical Andes. Conversely, when correcting the AMOC in the model, an increase in precipitation is observed (Fig. 4e).

Methods

Line 295: This is unclear. What does it mean for a glacier to be in or close to equilibrium with the climate? Does this mean that the glacier is in equilibrium with respect to its mass balance (not negative or positive)? If so I think it would help to say with respect to mass balance as "with the climate" is vague and could mean many different things.

Here we refer to glaciers in or close to equilibrium (mass balance = 0) with the climatic conditions. We changed the sentences accordingly.

Line 297: This can be clarified or deleted as it is sort of implied. Perhaps rephrase if it is thought that this sequence is necessary (but cross-cutting relationships seems pretty common knowledge) subsequent less extensive advances may leave additional moraines up-valley of older more extensive stillstands. When glaciers grow larger than prior extents, moraine sets within the valley may be destroyed and new younger moraines deposited in the maximum downvalley ice position.

We fully agree with R1 that this is common knowledge. But we believe it is important to point it out, as most of our analysis is based on this geomorphological statement.

Line 308: This is confusing - was ^{10}Be extracted and processed at two different labs? Or was the ^{10}Be from boulders processed in one lab and ^{10}Be and ^{14}C from "bedrock" processed from another? Also, how is bedrock collected from a glacier? Does this mean boulders or bedrock exposed when ice melted? Line 301 says ^{10}Be extraction occurred at CALM according to 50. Here the sentence talks about both being extract at Tulane. The next sentence (deleted) gives the ^{14}C extraction method ref but it seems out of place here given that both ^{10}Be and ^{14}C are mentioned in the prior sentence but only one method listed here. These first few sentences should be cleaned up and re-written for clarity.

You are right. Thank you. We rephrased this section and updated Table S1.

Line 317. Does this mean all new samples or all reported (some recalculated?) - clarify.
Yes – thank you. This has been clarified.

Line 318. erosion rates are provided on the tables....were these used in age calculation? Where are they derived from? Or are these the erosion rates that would be interpreted from the cosmogenic concentrations? Clarify.

We did not apply any erosion rate for the moraine boulders. Erosion rate was considered for the bedrock samples only. This has been clarified. We now write: ‘All $^{10}\text{Be}/^9\text{Be}$ isotope ratios were calibrated to the 07KNSTD⁵³ and new ^{10}Be moraine ages were determined using the CREP online calculator⁵⁵ with the LSD scaling model⁵⁶, assuming no denudation and using regional production rates. For the age of such samples, erosion has a very limited influence.

Line 322. What does this mean? Should "recalibrated" be "recalculated" here?

This was a mistake. It should have said ‘recalculated’ indeed. Thank you.

Line 325. This is not necessarily the most appropriate way to establish a moraine age. The paper referenced here describes a way to identify outliers. There are other publications about whether it is appropriate to use an average, a weighted average or the maximum moraine age. At least the authors might say why they decided to go with a weighted mean.

We do agree with the reviewer 1. There are different options to calculate a moraine age. Because selected sites are located in glacial valleys with possible influence of post depositional processes, we believe that calculating the mean from distinct samples on the same landform is more appropriate in this context (Jones et al., 2019). We used the weighted mean in order to better consider variations in analytic uncertainties between samples. Finally, we applied the same approach on a big dataset to obtain robust conclusions.

Line 327: Add information For bedrock samples - maybe include a sentence or two about the assumptions in sampling bedrock from valley floor: nuclide inventory affected by exposure, burial, and erosion which is a different set of assumptions than a sample of a boulder on a glacial moraine.

We now start this section with the following sentence: “Whereas moraine boulders typically experience a simple history with a single period of exposure and negligible erosion following deposition, proglacial bedrock samples from a valley floor may have undergone multiple episodes of exposure, burial by ice, and glacial erosion.”

Line 330. It would be helpful to read about a brief description of where the samples came from with respect to other landscape features - valley floor, valley wall, cirque, bedrock knob in valley, etc.

Such details are given on lines 370-375. We have now added some extra information on line 366.

Line 346. I don't follow this. It wasn't covered during the LGM but it was covered during the late glacial and LIA? Is this just for one of the samples (or just the samples that are green circles where only ^{10}Be ages are given?)

Indeed, this was really unclear. We apologize for this confusion. Details are now given on lines 370-385. A new Fig. S8 was also added.

Line 345. Perhaps clarify the difference between a cosmogenic concentration in a boulder on a moraine and a concentration from a bedrock sample. You treat them differently (appropriately) but the distinction isn't clear here or on the tables. For example, there is erosion occurring on the moraines and boulders....not because of burial by a glacier but since deposition through exposure. So this erosion rate is also modifying the nuclide inventory (albeit minimally) and you do have to either assume zero erosion or assume an erosion rate to this boulder to calculate the age - clarify this.

As mentioned above, we now clarify that exposure/erosion histories are likely different between boulder and proglacial bedrock samples. We now also note in the laboratory analysis section of the methods that we assume there was zero boulder erosion (Table S1).

Line 358. Why does this figure just show erosion rate and nothing about age? It would be helpful to see the potential scenario (exposure/burial) with a figure.

Thank you for this suggestion. We have updated the figure to show the exposure histories and corresponding erosion rates for each scenario.

Line 375-376: Was muogenic production considered in the model?

Yes, production by muons was included, as mentioned on lines 386-390 under the numerical model description.

Line 383: how do we know it is negligible

^{10}Be production under only 10 m of ice is ~1% that of surface production. Therefore, even 10,000 years of production in this case would yield only ~100 years worth of ^{10}Be . This is much lower than the apparent ages of P12 and P13 (~2000 years for ^{10}Be), and so would be only a minor contributor to the signal in these samples.

Line 388-391. This phrasing is confusing. Why is the "but" here? The first part of the sentence was saying it was ice-free for a certain period - isn't the idea that erosion occurs during glacial cover not during the ice-free period. I think the "but" should be "and" or "or"- the difference in concentrations applies a complicated history that may involve prior exposure or inherited nuclide concentrations (due to lack of complete erosion), or both.

Done

Line 430-431. Well, this introduces a bias....of course if you only look at records where there is only an early and late Holocene recorded you'll only see evidence for advances in the early and late Holocene. For example, in the northern Andes there are records that should be considered here. The story is still similar that there is an early Holocene advance, a mid-Holocene retreat/still stand, and a late Holocene advance, but a) in some cases there are moraines from the mid-Holocene, and b) the late Holocene moraines are well up-valley from the early Holocene moraines (which is quite different that those presented here). I recommend adding some studies (or at least a review study (i.e. Rodbell et al., 2009) to expand the tropical record in a more complete way. Bolivia is quite different than the northern Andes in important ways that are being overlooked here. I think this will add to the story.

This is a very important comment and we now include additional explanations about the criteria that were used to select the glacier records. We already addressed this issue and justified these criteria earlier in our response. Please see our response at the beginning of the document. We therefore only summarize our strategy in a few sentences here. Our goal is to identify when the largest Holocene glacial advance occurred. To really demonstrate that the largest glacier advance occurred during a specific period of the Holocene one has to prove that moraines from the other periods do not exist, or are located in an upper sector of the catchment (based on the well-known and accepted process of obliteration). Rodbell et al. (2009) mention glacial advances during the mid-Holocene based on minimum or maximum radiocarbon ages. Here we selected moraines dated by CRE ages, because any minimum or maximum age of a moraine would not be informative enough in our case. Existing CRE moraine chronologies in the Peruvian Andes do not show any mid-Holocene advance. We assume that glaciers may have advanced during the mid-Holocene due to local climatic conditions, but this advance was obliterated by the one occurring during the Late Holocene or the LIA, which was larger. Therefore, as we focus on the glacier evolution at a millennial time scale, we conclude that the mid-Holocene does not feature the largest extent in such an example. We understand that Figure 1 would be different with annual and continuous records, but minimum or maximum ages cannot be included to draw such a figure. We thus clarified the title of Figure 1, explaining that it is exclusively based on CRE moraine dating.

Line 446: Yes and this is extremely well established by many many many prior studies all over the world - the late glacial advances are related to YD, ACR, etc...seems that this is more background info rather than an interpretation. Not sure of the usefulness of this information here.

Ok - thank you.

Line 448: how us on a figure - refer to that figure here.

We deleted this paragraph which was no longer relevant for our purpose

Line 458. Does this mean for modern glaciers? For what period does this relationship hold because in the prior sentence it says the opposite and the LIA wasn't that long ago.

Yes – we meant modern glaciers. We rephrased this sentence. Thank you.

Line 460. This doesn't make sense - or it's redundant: the moraines were formed during periods of positive mass balance. So, the CRE moraine record and mass balance observations represent the same thing - no need to separate them here. This paragraph needs to be rewritten. I think the authors are trying to say that the current relationship between PDO and glacier mass balance doesn't hold for the LIA - it seems other SST proxies should be used here as well, also paleoENSO records (such as from Pacific marine records might be useful).

R1 is correct – the current relationship “does not hold for the LIA”. We rephrased this sentence and see our comment made earlier about ENSO and PDO. Paleo-ENSO records also are not consistent with Holocene glacier reconstructions (see Fig. 2). We now write: ‘Observations on tropical Andean glaciers conducted over the last decades show a positive relationship between mass balance and cold SSTs and *vice-versa*. This relationship is inconsistent with the PDO changes during the LIA.’

Line 482-483. Which model.

Done. We now write: ‘Statistical model presented in the next section

Line 485. Is this referring to the same model in the prior sentence? Or to the hosing experiments?

It refers to the hosing experiments. This has been clarified.

Line 488 What does this mean here?

Here “In line” means in agreement with. We changed the sentence to clarify this.

Line 492 scores? what does this mean? Fit?

Yes - correlation value. We changed the sentence to clarify this.

Lines 505-507. The sentence in 505 seems at odds to that in 506-507 - on one hand they are saying the simulations are capturing the fingerprint in terms of climate (which I assume means reproducing a climate suggested by many proxy records) but at the same time they say are not capturing a main component (precipitation) according to proxy data - or are they saying in most places of the NA it is capturing the main components but not in the Alps?

Indeed, this was unclear. Thank you - we have rephrased this and now write: ‘These simulations are thus capturing the main AMOC fingerprints. We notice that precipitation changes over the Alps associated with a weakening of the AMOC are not significant, despite evidence of enhanced precipitation documented from proxy records’ (see lines 566-569 in the revised version).

Figures

Fig 2. Why Scandinavia only? Would help to have latitude and to see this plot for ~60N, 16S and 50S or something.

We believe there is a misunderstanding. This curve was produced by ref 25 and corresponds to the mean of 16 glaciers in Scandinavia only.

Fig 2. Curve 2. I don't see the connection between the glacier length and ice free conditions at ice cap without showing the position of the ice cap in other times (late and early Holocene).

Again, we believe there is a misunderstanding. The different symbols show moraine chronologies for different glaciers, including Quelccaya ice cap. On the other hand, independent ^{14}C ages from plant remains and *in-situ* ^{14}C CRE ages show that the Quelccaya ice cap was small for a period of several centuries in the mid-Holocene. The fact that no chronology shows a moraine dated to the mid-Holocene suggests that glaciers were larger during the late Holocene than the mid-Holocene (eroding previous mid-Holocene moraines). Combining these data with *in situ* ^{14}C ages published in this study or from previous studies in Europe and Greenland enabled us to draw this synthetic curve that shows the evolution of the glacier size in TANAR.

Fig 3. Give each plot above a title with text ON the figure. Put units and other important info on the scale bars.

We have tried to make the legend more easily understandable. Fig 3a shows the AMOC trend over the last century based on ref 30. Fig. 3b shows the sea surface temperature (SST) at global scale derived from Fig 3a. and Fig. 3c shows the correlation between mass balance and SSTs from Fig. 3b.

Fig C. This is confusing because nothing on this plot clearly shows mass balance. This plot is a map showing the spatial distribution of what...there are not units given, just colors. I infer that these colors represent average SSTs between Oct-March over the period 1974-2016. But instead is it some combination of SST and mass balance? Clarify here in the caption and include units.

We indicated in the Figure caption that this Figure shows the ‘Spatial correlation between October-March mass balance on Bolivian Zongo glacier (white dot) and global SSTs over the period 1974-2016. Both mass balance and SST data were smoothed with a 9-year running mean.’ There are no units shown, because correlation coefficients are unitless. Hence the plot shows how SST anomalies, anywhere in the global ocean correlate with mass balance on Zongo glacier in the months of October-March. For example, locations that show positive correlation coefficients feature SST that are anomalously warm (cold) when the mass balance is anomalously positive (negative) on Zongo glacier. This is a fairly standard and straightforward analysis and we feel the caption adequately explain the content of the Figure.

How is this figure related to S6 (both are referred to in the main text at the same location). The captions are very similar except S6 says "near surface temperature". S6 has slightly different colors but similarly does not have any units or explanation about what the colors mean.

Fig. S6 shows the same analysis with near-surface air temperature, except that the data was low-pass filtered first to remove interannual variability (in order to consider relationships at the decadal scale). This was indicated in the supplementary text. However we removed this figure from the new version.

Also, why focus on the Zongo glacier here? What is the significance. This is not explained in the text or supplement or captions.

Because this is the longest existing glacier mass balance time series from this region. This has now been clarified in the supplementary text.

Fig 4. It seems this zero line intersects with 8ka and 4ka quite nicely...is this an attempt to set a threshold for glacial advance? If precip is below 0mm, neg mass balance, etc.? I think that would be helpful to show (similar with my comments on S5).

This is a very interesting comment, which we agree with. However, we believe our semi-empirical model is too simplistic to make such bold claims.

Fig 4a: What are the units of the scale bar? Since it is a difference, does blue indicate colder temps during hosing compared to control?

Indeed, blue colors indicate colder temperatures than those found in the control run. We have clarified this aspect in the revised manuscript.

Fig 4cd If both are shown, why does label on graph just say TraCE?

This is a misunderstanding. Fig. 4c shows the results for TraCE only, while Fig. 4d shows the same analysis for LOVECLIM only. We separated these analyses into two separate figures to improve readability.

Fig 4a: What are the units of the scale bar? Since it is a difference, does blue indicate colder temps during hosing compared to control?

Indeed, colder than the control run. We indicated it.

Fig 4cd If both are shown, why does label on graph just say TraCE?

No, there is a misunderstanding. See our response above. Fig. 4c shows the results for TraCE only, while Fig. 4d shows the same analysis for LOVECLIM only to make the figure easier to read.

Fig S2. The map has the Green ages as ^{14}C (for example P12, P13) while the color here is green. I don't understand the color scheme here. If the circles are yellow they are ^{14}C ages AND ^{10}Be ages while the circles that are green are only ^{10}Be ages.

Yes, the yellow circles correspond to samples for which both ^{10}Be and ^{14}C ages were measured. P12 and P13 are represented by green circles, as we did not measure the ^{14}C . These samples correspond to bedrock and therefore the color of the circle is different from moraine boulders.

Fig S5. So it seems from 8-4 ka (no ice advances except maybe in a few Andean valleys around 5-6ka, mid-valley): higher intensity AMOC (a) above a threshold of $\sim 1\text{Sv}$ and lower precipitation (below zero). Temperature not clear - but precip is way higher during 4-0 when temps are higher.

Yes, we agree with this general assessment. However, our semi-empirical statistical model is rather simplistic so this may stretch the interpretation too far. One also has to consider the differences between the simulations from the two GCMs.

Fig S5. So it seems from 8-4 ka (no ice advances except maybe in a few Andean valleys around 5-6ka, mid-valley): higher intensity AMOC (a) above a threshold of $\sim 1\text{Sv}$ and lower precipitation (below zero). Temperature not clear - but precip is way higher during 4-0 when temps are higher.

Yes, we agree. This is a possible scenario, but because our semi-empirical statistical model is simple, this may be an over interpretation. One has also to consider the distinct patterns resulting from the simulations of the two GCMs.

Aren't they from both TraCE and LOVECLIM since there is both a black and a blue line as indicated in (a)?

This is true for Figure (a), but this Figure shows the Sv evolution. In Figure (b) it is AMOC-corrected precipitation from TraCE only and in (c) the same variable, but for LOVECLIM only.

Reviewer 2

1- I agree with the author that glacier primarily responses to a temperature change in summer. However, AMOC variability primarily manifest temperature anomaly in winter. The author should compare the

summer temperature anomaly related to the AMOC variability with that induced by insolation changes. I believe that insolation has more pronounced effect than the AMOC does.

We thank R2 for this remark. Indeed, we assume that glaciers in the Atlantic basin of the northern Hemisphere were mostly driven by summer temperature. However, this paper focuses on glaciers from different regions. For that reason, we distinguished in our response the influence of the AMOC on tropical glaciers from its impacts on glaciers located in the mid-latitudes of the northern hemisphere.

Tropical glaciers are mostly controlled by precipitation changes from September to March (Favier et al., 2004). Holocene precipitation changes from TraCE and LOVECLIM models in full-forcing experiments are inconsistent with the observed glacier changes, unless we invoke the (corrected) AMOC impacts (Fig. 4e). Consequently, we claim that the AMOC is the key driver in this region. As shown in Fig. 4e, AMOC-corrected precipitation changes are characterized by dry conditions during the mid-Holocene, in agreement with proxy records (Stansell et al., 2017). This trend corroborates the large glacier shrinkage documented in this paper. In addition, wetter and cooler conditions during the late Holocene compensate for increasing insolation changes that favor glacier readvance (Fig. 4e).

As far as impacts of the AMOC on northern hemisphere mid-latitude glaciers are concerned, we have followed the recommendation of R2. We have analyzed the summer temperature changes from both TraCE and LOVECLIM full forcing experiments, including AMOC changes. Results from Figure 1r (see below) show that summer temperatures from the full forcing experiments are consistent with moraine records as reported in this paper, with two temperature minima during the early and late Holocene, interrupted by a warmer period during the mid-Holocene. In other words, cool temperatures during the Late Holocene would not result from AMOC forcing but from a combination of different forcings, including a decrease in summer insolation (but not exclusively, see our point 3 below and associated Fig. 3r). Hence, while our AMOC correction approach does not contradict moraine records, this correction may not be fully necessary (Fig. 1r).

Nevertheless, there are additional arguments, depicted below, that suggest that the role of the AMOC was not negligible in the high latitudes either, while it was clearly central to tropical glaciers' evolution:

Fig. 1r. Boreal summer temperature anomaly in the northern mid-latitudes from the full forcing TraCE experiment (black) and AMOC-corrected temperature (blue).

- 1- We notice a major discrepancy between AMOC changes in the two models and the recent reconstructions (Fig. 2; Fig. S6 shown in the main text) at an annual time scale. In the two models the AMOC strength is progressively increasing, while observations reveal contrasting AMOC changes during the mid and Late Holocene. Consequently, in the two models the AMOC is too weak during the mid-Holocene and too strong during the Late Holocene.
- 2- We also notice a strong discrepancy in summer temperature between results from the two full forcing experiments and the most recent reconstruction of summer temperature (Kaufman et al., 2020). Summer temperatures from TraCE, for instance, are much cooler than proxy reconstructions during the Late Holocene (Fig. 2r). This difference may be related to a bias in the proxies used in the reconstructions (Bova et al., 2021) or due to errors in the model experiments. Following the second option related to errors in the model experiments, the bias in temperature during the Late Holocene could explain why our correction of simulated AMOC summer temperature changes is apparently not necessary for the Late Holocene in the high latitudes of the northern hemisphere.

Fig. 2r. Comparison between boreal summer temperature anomalies in the northern mid- and high latitudes from TraCE full forcing experiment (black curve) and the most recent reconstruction from Kaufman et al. (2020).

- 3- Summer temperature changes in the high latitudes of the northern hemisphere from TraCE and LOVECLIM experiments using insolation changes only reveal a monotonic decrease with a minimum value during the Little Ice Age (Fig. 3r). This trend is inconsistent with moraine records reported in this paper, as the Little Ice Age does not correspond to the maximum extent of the mountain glaciers during the Holocene. Consequently, insolation cannot be the single driver of glacier changes in the mid-high latitudes of the northern hemisphere.

Fig. 3r. Boreal summer temperature in Greenland from TraCE using full forcing in red and insolation only in black.

- 4- LOVECLIM experiments show significant differences in summer temperature between 30-60°N and 60-90°N (Kobashi et al., 2017). These regional differences are much reduced when temperature is AMOC-corrected, indicating they are in line with glacier observations (TANAR trend).
- 5- At an annual timescale the correlation between temperature changes from TraCE or from LOVECLIM with reconstructions provided by Marcott et al. (2013) or by Kaufman et al. (2020) is improved once we apply our semi-empirical model of AMOC correction.
- 6- We show that a stronger AMOC during the mid-Holocene is a key driver of atmospheric temperature changes, even in summer in the northern hemisphere (cf. Fig. R1), which play a significant role in glacier retreat at that time, in line with recent SST reconstructions (Allan et al., 2021).

Consequently, considering the different points listed above, we prefer considering temperature changes at the annual timescale. However, we have revised the discussion related to AMOC impacts on temperature changes in the mid-latitude of the northern Hemisphere throughout the document.

Finally, we would like to recall the strong influence of the AMOC on regional climatic conditions in the northern hemisphere which is highlighted in this paper based on both data and model results. Our results confirm recent TraCE experiments (Renssen et al., 2009; Mc Kay et al., 2018) which documented the strong AMOC influence on northern hemisphere temperature changes during the early Holocene when combined with ice sheet remnants. Here we show that AMOC changes during the mid and Late Holocene have a significant influence on glaciers in TANAR. Our analysis also corroborates other studies conducted in western Greenland (O'Hara et al., 2017) that suggested a major role of the weakening of the sub-polar gyre on glacier change. In addition, Allan et al. (2021) used a SST-SSS (sea surface temperature and salinity) multi-proxy reconstruction to show that after ~5 ka BP a decrease in phototrophic taxa marked a two-step cooling of surface waters. The first cooling period started at ~5 ka BP, and the second one at ~3 ka BP, with a shift toward colder conditions and higher SSS, in phase with AMOC changes shown in our Figure 2. Based on these data, the authors concluded that a strong interaction exists between the changes in ocean water masses and ice-margin history of the GrIS and glaciers during the Holocene.

2. The glacier has long response time to temperature change, the equilibrium retreat of glacier in response to warmer climate can be thousands of years. Therefore, the minimum extent of glacier will of course take place later than the temperature peak (e.g., Briner et al., 2016). The shape difference of insolation and glacier extent can also be induced by such delay response. In other words, the advance/retreat of ice sheet extent does not directly depend on the cooling/warming of the climate, but on the background temperature which is below/above the threshold to cause the advance/retreat of ice sheet extent. In my opinion, though the NH summer insolation was reducing from early to mid Holocene, the temperature might still stay in a range leading to retreat of glacier. The threshold was reached at around mid to late Holocene, from then the ice sheet begin to re-advance. Therefore there is a delaying effect of the peak-insolation or temperature on the occurrence of minimum ice sheet extent. The author should at least consider this delaying effect when examining the affecting factors of ice sheet changes.

We fully agree with this comment, except that we did not focus on ice sheet changes but on mountain glaciers. We estimate the response time of the investigated glaciers in this paper to be less than 150 y (i.e. lower than the ¹⁰Be uncertainties for the early Holocene). This estimate is based on the paper of Raper and Braithwaite (2009), who applied a glacier modeling approach to determine the response time of 14,327 glaciers located in seven regions, including the European Alps, the Arctic and New Zealand. Their results show that the response time of mountain glaciers is less than 75 y for a glaciated area of 10 km², which is larger than the size of the investigated glaciers in this paper. Raper and Braithwaite (2009) do not provide a response time for Greenlandic mountain glaciers, but show different values for Arctic regions located at different latitudes. According to Raper and Braithwaite (2009) glaciers with a surface area of 10 km² located at a latitude of 70°N have a response time of 155 years. The glacier selected in this paper is located at Zackenberg, 71°N and is smaller than 3 km². We thus assume that 155 y is a conservative estimate for the selected glacier in this paper. There are no data for the tropical Andes but direct observations of the front variations and climate setting on Charquini glaciers which are used in this paper, show a response time of less than 10 years (Rabatel et al., 2013). Based on this short response time of mountain glaciers and the fact that our analyses are conducted at a millennium time scale, we argue that moraines dated by cosmogenic ages reflect the investigated AMOC forcing.

References used in this response

- Allan et al. (2021). Insolation vs. meltwater control of productivity and sea surface conditions off SW Greenland during the Holocene. *Boreas*. <https://doi.org/10.1111/bor.12514>. ISSN 0300- 9483.
- Benson et al. (2007). Surface-exposure ages of Front Range moraines that may have formed during the Younger Dryas, 8.2 cal ka, and Little Ice Age events. *QSR*, 26, 1638–1649.
- Bova et al. (2021). Seasonal origin of the thermal maxima at the Holocene and the last interglacial *Nature*, 589, 548–553.
- Drijfhout et al. (2013). Spontaneous abrupt climate change due to an atmospheric blocking–sea-ice–ocean feedback in an unforced climate model simulation. *PNAS* 110, 19713–19718.
- Favier, V., Wagnon, P., Ribstein, P. (2004). Glaciers of the outer and inner tropics: a different behavior but a common response to climatic forcing. *Geophys. Res. Lett* 31, doi: 10.1029/2004GL020654
- Jones et al. (2019). iceTEA: Tools for plotting and analysing cosmogenic-nuclide surface-exposure data from former ice margins. *Quaternary Geochronology*, 51, 72-86.

- Kaufman et al. (2020). A global database of Holocene paleotemperature records. *Scientific data*. doi.org/10.1038/s41597-020-0445-3
- Kobashi et al. (2017). Volcanic influence on centennial to millennial Holocene Greenland temperature change. *Scientific Reports*, **7**, 1444: DOI:10.1038/s41598-017-01451-7
- Licciardi et al. (2009). Holocene glacier fluctuations in the Peruvian Andes indicate northern climate linkages. *Science*, **325** (5948), 1677–1679.
- McKay et al. (2018). The Onset and Rate of Holocene Neoglacial Cooling in the Arctic. *GRL* **45**, (2018) doi.org/10.1029/2018GL079773
- Marcott et al. (2013). A reconstruction of regional and global temperature for the past 11,300 years. *Science* **339**, 1198–1201.
- Marcott et al. (2019). ^{10}Be age constraints on latest Pleistocene and Holocene cirque glaciation across the western United States. *Npj Clim Atmos Sci* **2**, **5**, doi.org/10.1038/s41612-019-0062-z
- O'Hara et al. (2017). A ^{10}Be chronology of early Holocene local glacier moraines in central West Greenland. *Boreas* **46**, 655–666.
- Rabatel et al. 2013. Current state of glaciers in the tropical Andes: a multi-century perspective on glacier evolution and climate change. *The Cryosphere*, **7**, 81-102.
- Raper S.C.B., and Braithwaite R. J. (2009). Glacier volume response time and its links to climate and topography based on a conceptual model of glacier hypsometry. *The Cryosphere*, **3**, 183–194.
- Renssen et al. (2009). The spatial and temporal complexity of the Holocene thermal maximum. *Nature Geoscience*, **2**, 411-414.
- Rodbell, D. T., Smith, J. A., Mark, B. G. (2009). Glaciation in the Andes during the Lateglacial and Holocene. *QSR* **28**, 2165-2212.
- Solomina et al. (2015). Holocene glacier fluctuations. *QSR*, **111**, 9-34.
- Stansell et al. (2014). Proglacial lake sediment records reveal Holocene climate change in the Venezuelan Andes. *QSR* **89**, 44-55.

Reviewers' Comments:

Reviewer #1:

Remarks to the Author:

Overall, the paper is much improved by the changes the authors have made. I appreciate their detailed and specific responses to my earlier comments and edits. The results and interpretation are compelling. I think this paper is acceptable for publication pending some minor revisions as there are minor elements of the text that remain unclear or difficult to understand as written. The criteria provided for how glacier catchments were selected is helpful, however I have a few concerns that are likely just a misunderstanding arising from slightly ambiguous text. This is important because their entire dataset and description of the TANAR-pattern rest on their data selection process. I am most confused by the first two criteria provided, so I'll discuss those first. I then provide comments where there was still some ambiguity in the text. Overall, I am excited to see this work published and am appreciative of the authors thoroughness in addressing many of my initial comments and concerns.

Lines 493-495: 1) We did not consider catchments with moraines that may correspond to the early Holocene or an older period such as the Younger Dryas if uncertainties are too large to distinguish between YD and Early Holocene.

Do the authors define the early Holocene as 10-8ka? Or 11.6-9ka? The Holocene begins at the end of the Younger Dryas, 11,700ka, and given that all of the dating methods used have about a 5-10% uncertainty, it is definitely possible for a ~11ka moraine (technically early Holocene) to have an error that could make it 11,700ka. Perhaps the authors can articulate that they are only considering the early Holocene to be 10-8ka (if this is true) and they did not consider any moraines that have errors falling outside of this range? Although I think this is limiting.

Lines 496-501: 2) We did not select catchments with undated moraines in-between the LIA and late glacial moraines. If another superimposed forcing factor played a significant role, pre-LIA moraines may have formed. Following this strategy, catchments with obvious undated moraines immediately downslope from the LIA front position were ignored (as these undated moraines may have been formed either during the mid-Holocene or during the early Holocene and thus may reflect the influence of other forcings).

The way this reads it seems they are looking for a pattern and ignoring any catchments that don't fit the pattern....which seems like it would bias their results. Or did they establish the TANAR-pattern from THEIR new ages and then went to the Ice-D database looking for matches? I guess it seems they should consider all of the data and then look for a pattern rather than the other way around. I do understand the need to limit variables. But this needs to be more clearly stated and explicit. In their rebuttal they state: "To do so, our approach entails testing different possible forcings without any prior assumptions." Yet it seems like this is exactly what they are doing with criteria 2. Their prior assumptions are that if there are moraines between the early Holocene and LIA moraines then there are other forcings involved so they did not include those catchments...I must be missing something here. If the TANAR-pattern is, as they state in Figure 1 caption: "a larger extent during the early Holocene than during the Late Holocene with substantial shrinking during the mid-Holocene" and they have rejected all catchments with undated moraines outboard (down-valley) of LIA, then they might be missing mid-Holocene moraine advances....although there still might have been shrinking as they were smaller than the early Holocene moraines.

Line 52: "recording their maximum size" – this is still confusing. Maximum during the Holocene (not as large as Pleistocene extents) and also only considering small mountain glaciers, right? For example, in the eastern Sierra Nevada of CA the early Holocene moraines are larger than LIA moraines...

Lines 81-87: sometimes TANAR is preceded by "the" and sometimes it is not – be consistent

throughout. Minor grammatical issues in added text "on millennial time scale" – I think it should be "on a millennial timescale" or "on millennial timescales". Instead of "to be a major driver" should be "are a major driver". I still think it would be useful here to say EXACTLY what the TANAR behavior is. The authors say in their rebuttal: "When the largest advance of a glacier occurred in the early Holocene and the second largest during the Holocene including the LIA, we assume it corresponds to the TANAR pattern." in response to the Line 140 comment. This, or something like this, should go here in lines 81-87 to be very clear. I think they mean the second largest during the LATE Holocene including the LIA, but the word "late" was not included in their sentence 0 so do they mean the entire Holocene?

Line 87: "to support the AMOC-influence hypothesis": do they mean instead "to test"?

Lines 114-117: I think some of my confusion is the use of "maximum extent". The authors refer to the maximum extent for the early, mid, and late Holocene...but also, in other places talk about the maximum Holocene extent (furthest down valley during the last 11,700 years). I think somewhere it would be good to articulate when they mean early, mid, and late are here, since early seems to be 10-8, mid is 8-4 and late is 4-0, although sometime late is LIA. Figure 1 seems to articulate the periods as 11.6-9ka, 8-4, and 3-LIA. Be sure there is consistency throughout. I believe these lines are saying that the early Holocene is the furthest down-valley extent but that the LIA is the next furthest down-valley extent and it occurred during the LIA. Seems "Late Holocene" shouldn't be a capitalized L if early and mid are not.

Lines 135-137: Great summary – could be useful (or an abbreviated version) above. Still the language is confusing "early Holocene maximum extent" (do they mean the maximum for the whole Holocene or just for the early Holocene? In line 116 they use the same phrasing "late Holocene maximum extent" to mean just the maximum of the late Holocene not the entire Holocene.

Line 139: instead of "differences with TANAR", perhaps "differences compared to TANAR glaciers".

Line 141: instead of "the maximum extent" perhaps "the maximum Holocene extent".

Line 255-258 and S6: "In the transient simulations, following a minimum at 10-8 ka, the simulated AMOC slightly increases during the rest of the Holocene while the reconstructions suggest a maximum in the mid-Holocene, followed by a weakening towards the Late Holocene (Supplementary Fig.S6)." Perhaps use the same terms as is shown on the figure. Here's what I think the authors are saying: While the LOVECLIM and TraCE transient numerical simulations show a minimum around ~8ka followed by a steadily increasing AMOC through the rest of the Holocene, the reconstructed AMOC shows a similar low during the early Holocene but with a maximum in the mid-Holocene followed by a weakening towards the Late Holocene (Supplementary Fig.S6). In fact, it looks like the LOVECLIM is nearly opposite of the reconstructed AMOC at times (e.g. 6ka)...Sometimes the LOVECLIM and TraCE are called transient numerical simulations and sometimes called full forcing experiments...perhaps use the same language throughout. S6 b and c: suggest using another color than blue for the AMOC corrected as it is confusing compared to S6a where the blue and black represented something different.

Line 278: "in line with" is unclear. Perhaps "corresponding temporally with glacier advances. One thing is that while cooler temperatures and higher precipitation cause advance, the abandonment of the moraines is caused by retreat, so here where the cooler temperatures cause advance it is also important then that the temperature rises or precipitation decrease so that then the ice retreats and leaves a Late Holocene (or other) set of moraines.

Line 282: It isn't clear what "this" and "those" refer to exactly. I don't think this sentence is really necessary.

Fig 1: It seems that "AMOC pattern" should be "TANAR pattern" – it is confusing if they are the same or different. Just pick one name for the pattern and stick to it (or are they different?) It seems the AMOC pattern here is described in the caption as: "a larger extent during the early Holocene than during the Late Holocene with substantial shrinking during the mid-Holocene." Which is similar to what they described in their rebuttal as the TANAR pattern (see comment above for lines 81-87. In their rebuttal they say: "As their evolution shows a good correspondence with AMOC changes, we replaced TANAR pattern in Fig. 1 with AMOC pattern.", but I think this is confusing – maybe just SAY that in the caption but refer everywhere to the pattern as the TANAR pattern.

Fig 2: Hopefully all of the figures will be larger – it is too small to read at this scale.

Fig 3: Suggest making numbers within circles bold for easier reading.

Reviewer #2:

Remarks to the Author:

Please see my attachment.

Review report on 'AMOC control on millennial-scale glacier changes during the Holocene'

The authors describe a V shape of TANAR glacier changes during the Holocene, which is similar to the shape of AMOC evolution. They claim that the AMOC related temperature and precipitation change has controls the millennial-scale glacier evolution over TANAR. The study is interesting. And, I know that reconstruction of past climate and glaciers are difficult and require hard working. However, I was still not convinced by the results.

The authors have argued that the temperature evolution over the mid-latitude of Northern Hemisphere is the primary driver of the glaciers. My major concern is that the AMOC variations have minor impact on the Holocene temperature evolution of the Northern Hemisphere if compare with the external insolation change. The Holocene summer insolation anomaly is on the order of 40 Watt/m², which is very, very strong in controlling the general shape of the summer temperature evolution, noticing that the strongest CO₂ emission projection (RCP8.5) can only produce 8.5 Watt/m² radiation anomaly.

As also shown in Fig. 1r, the external forced temperature evolution is on the order of 1.5 degrees, whereas, the estimated impact of AMOC is on the order of only 0.5 degree. Moreover, insolation change, in particularly, the precession change can largely affect the position off ITCZ, as Holocene precession anomaly could lead to warming the Northern Hemisphere but cooling the south.

Fig. 1r. Boreal summer temperature anomaly in the northern mid-latitudes from the full forcing TraCE experiment (black) and AMOC-corrected temperature (blue).

If AMOC do has a systemic impact on the glaciers. It must be first detected in the temperature signal, rather than the glaciers. However, we did not observe any such variations during the Holocene. Dramatic AMOC change, such as Heinrich events do have strong impact on the climate, particular during the last glacial period. However, during the Holocene, most of the temperature proxies does not show a fingerprint of AMOC changes. The reconstructed temperature evolution fit well with the insolation change.

In my opinion, the external forcing is the major driver of the glacier evolution. As the background temperature evolution also show an upside-down V pattern. The more extensive glaciers during the early Holocene may related to relatively low temperature results from the residual Laurentide ice sheet and low CO₂.

The minimum glaciers during the mid-Holocene can be results of melt of Laurentide ice sheet and increased summer insolation. And finally, the re-advance of the glacial during the late Holocene is forced by decrease summer insolation.

Overall, I found the results show here is not robust.

Minor comments:

1. Line 229: Summer in the 16S is DJF, but 60N is JJA. I suggest to use month to indicate the insolation.
2. The authors compared the TANAR glaciers with other glaciers, which are only 3 glaciers. Two of them are over the Southern Hemisphere, which of course will have different patterns, because the insolation evolution is different from the Northern Hemisphere. If the AMOC has an impact over the Northern Hemisphere, it should have similar effect on the Southern Hemisphere, because AMOC is features a temperature seesaw between different Hemispheres.
3. Line 871: Specify which season does the temperature comes from? JJA or DJF or Annual mean?

Reviewer #1 (Remarks to the Author):

Overall, the paper is much improved by the changes the authors have made. I appreciate their detailed and specific responses to my earlier comments and edits. The results and interpretation are compelling. I think this paper is acceptable for publication pending some minor revisions as there are minor elements of the text that remain unclear or difficult to understand as written. The criteria provided for how glacier catchments were selected is helpful, however I have a few concerns that are likely just a misunderstanding arising from slightly ambiguous text. This is important because their entire dataset and description of the TANAR-pattern rest on their data selection process. I am most confused by the first two criteria provided, so I'll discuss those first. I then provide comments where there was still some ambiguity in the text. Overall, I am excited to see this work published and am appreciative of the authors thoroughness in addressing many of my initial comments and concerns.

Lines 493-495: 1) We did not consider catchments with moraines that may correspond to the early Holocene or an older period such as the Younger Dryas if uncertainties are too large to distinguish between YD and Early Holocene.

Do the authors define the early Holocene as 10-8ka? Or 11.6-9ka? The Holocene begins at the end of the Younger Dryas, 11,700ka, and given that all of the dating methods used have about a 5-10% uncertainty, it is definitely possible for a ~11ka moraine (technically early Holocene) to have an error that could make it 11,700ka. Perhaps the authors can articulate that they are only considering the early Holocene to be 10-8ka (if this is true) and they did not consider any moraines that have errors falling outside of this range? Although I think this is limiting.

We confirm that we considered the early Holocene as 11.6-9 ka as reported in Fig. 1 which corresponds to the classical definition. We also confirm that glaciers with early Holocene moraines that would overlap with the Younger Dryas period were not considered. We also confirm that we included in our analysis glaciers whose behavior is inconsistent with AMOC changes. On Fig. 1 for instance glaciers located in the mid latitude of the southern hemisphere or glacier in Himalaya do not follow AMOC trend (See Fig 1). To summarize we distinguished three categories: glaciers whose behavior is consistent with observed AMOC changes, those with behavior that is not consistent, and finally glaciers for which their change is ambiguous. This ambiguity is related to two possibilities:

- a) **The evolution of the glacier during the early Holocene is not analyzed**
- b) **The chronology of the early Holocene remains uncertain and may be attributed either to the Younger Dryas or the mid-Holocene due to dating uncertainties.**

Glaciers belonging to the first two categories (consistent or inconsistent AMOC behavior) were considered in our analysis, while uncertain chronologies were not. We included the following explanation in the method section, which we hope clarifies our strategy.

Lines 496-501: 2) We did not select catchments with undated moraines in-between the LIA and late glacial moraines. If another superimposed forcing factor played a significant role, pre-LIA moraines may have formed. Following this strategy, catchments with obvious undated moraines immediately downslope from the LIA front position were ignored (as these undated moraines may have been formed either during the mid-Holocene or during the early Holocene and thus may reflect the influence of other forcings).The way this reads it seems they are looking for a pattern and ignoring any catchments that don't fit the pattern....which seems like it would bias their results.

We understand this concern, but this is of course a misinterpretation. As expressed in the previous paragraph, we rejected ambiguous chronologies but included glaciers even if their behavior is inconsistent with the AMOC pattern, as observed in several regions, notably those located in the Southern Hemisphere mid latitudes or in the Himalayas.

Or did they establish the TANAR-pattern from THEIR new ages and then went to the Ice-D database looking for matches? I guess it seems they should consider all of the data and then look for a pattern rather than the other way around.

Again, we apologize for such an unclear description of our approach. We agree with the first part of the sentence they establish the TANAR-pattern from THEIR new ages and then went to the Ice-D database looking for matches? but disagree with the last three words of it. Yes, we established the TANAR-pattern based on our new ages and then went to the Ice-D database. However, we did not exclusively select glaciers in ICE-D that are consistent with the TANAR pattern. As expressed earlier, we also included glaciers that are characterized by a behavior that is inconsistent with our dataset. Figure 1 clearly shows that there are glaciers and regions that do not follow the TANAR pattern. Moreover, previous recommendations made by R1 to explore regions outside of the Atlantic domain are particularly relevant for this purpose.

I do understand the need to limit variables. But this needs to be more clearly stated and explicit. In their rebuttal they state: “To do so, our approach entails testing different possible forcings without any prior assumptions.” Yet it seems like this is exactly what they are doing with criteria 2. Their prior assumptions are that if there are moraines between the early Holocene and LIA moraines then there are other forcings involved so they did not include those catchments...I must be missing something here.

Indeed the term “include” was unclear and not at all a good choice! We apologize and are grateful for the thorough review of this aspect, which is very helpful. These catchments were included in our analysis and included in Figure 1. “Included” was a bad choice to express the idea of being “coherent with” the TANAR pattern. We removed this sentence.

If the TANAR-pattern is, as they state in Figure 1 caption: “a larger extent during the early Holocene than during the Late Holocene with substantial shrinking during the mid-Holocene” and they have rejected all catchments with undated moraines outboard (down-valley) of LIA, then they might be missing mid-Holocene moraine advances...although there still might have been shrinking as they were smaller than the early Holocene moraines.

Yes, we agree with this assessment. These moraines may indeed stem from the mid-Holocene. However, these moraines may also be older than the mid-Holocene. Note that glaciers with undated moraines in between the late glacial-early Holocene transition and the LIA are very rare in the tropics. Altogether these undetermined cases represent about 18% of our dataset. Even if this percentage seems high, it does not influence our main conclusion, as most cases are not located in the TANAR, but in the Himalayas (Saha et al., 2018) with 6 cases corresponding to the two previous categories defined at the beginning of our response, and in New Zealand, (Kaplan et al 2010, 2013) with four glaciers that correspond to category a). The rest are located in Greenland, with three glaciers corresponding to category a) (OHara et al., 2017; Schweinsberg et al., 2019), Peru, with two glaciers corresponding to category a) (Hall et al., 2009) and one to category b), (Stroup et al., 2015), Alaska (category b; Young et al., 2009) and Africa (category a; Vickers et al., 2021).

Line 52: “recording their maximum size” – this is still confusing. Maximum during the Holocene (not

as large as Pleistocene extents) and also only considering small mountain glaciers, right? For example, in the eastern Sierra Nevada of CA the early Holocene moraines are larger than LIA moraines...

Exactly. Yes, we meant of course smaller than the LGM. Here, maximum size refers exclusively to the Holocene. This was indicated in one sentence in the introduction. We assumed that this is implicit as the papers discusses the Holocene exclusively. However we now added “during the Holocene”.

Lines 81-87: sometimes TANAR is preceded by “the” and sometimes it is not – be consistent throughout.

Yes, thank you: done.

Minor grammatical issues in added text “on millennial time scale” – I think it should be “on a millennial timescale” or “on millennial timescales”. Instead of “to be a major driver” should be “are a major driver”.

Done.

I still think it would be useful here to say EXACTLY what the TANAR behavior is. The authors say in their rebuttal: “When the largest advance of a glacier occurred in the early Holocene and the second largest during the Holocene including the LIA, we assume it corresponds to the TANAR pattern.” in response to the Line 140 comment. This, or something like this, should go here in lines 81-87 to be very clear.

You are right. Thank you, we have corrected this mistake.

I think they mean the second largest during the LATE Holocene including the LIA, but the word “late” was not included in their sentence 0 so do they mean the entire Holocene?

You are right. Thank you, we indicated the Late Holocene.

Line 87: “to support the AMOC-influence hypothesis”: do they mean instead “to test”?

Yes, this has been changed.

Lines 114-117: I think some of my confusion is the use of “maximum extent”. The authors refer to the maximum extent for the early, mid, and late Holocene...but also, in other places talk about the maximum Holocene extent (furthest down valley during the last 11,700 years). I think somewhere it would be good to articulate when they mean early, mid, and late are here, since early seems to be 10-8, mid is 8-4 and late is 4-0, although sometime late is LIA. Figure 1 seems to articulate the periods as 11.6-9ka, 8-4, and 3-LIA. Be sure there is consistency throughout. I believe these lines are saying that the early Holocene is the furthest down-valley extent but that the LIA is the next furthest down-valley extent and it occurred during the LIA. Seems “Late Holocene” shouldn’t be a capitalized L if early and mid are not.

We apologize for this unclear message. We confirm our analysis used indeed these periods 11.6-9ka, 8-4, and 3-LIA. If the glacier follows a TANAR pattern then it is characterized by a maximum Holocene extent in the early Holocene, followed by the second largest extent located down-valley associated with the late Holocene. Glaciers that do not follow a TANAR pattern may have reached their maximum Holocene extent during the mid-Holocene or during the LIA.

Lines 135-137: Great summary – could be useful (or an abbreviated version) above. Still the language is confusing “early Holocene maximum extent” (do they mean the maximum for the whole Holocene or just for the early Holocene?)

Here we refer to the whole Holocene. We clarified this.

In line 116 they use the same phrasing “late Holocene maximum extent” to mean just the maximum of the late Holocene not the entire Holocene.

Done

Line 139: instead of “differences with TANAR”, perhaps “differences compared to TANAR glaciers”.

OK, thank you – done.

Line 141: instead of “the maximum extent” perhaps “the maximum Holocene extent”.

Done.

Line 255-258 and S6: “In the transient simulations, following a minimum at 10-8 ka, the simulated AMOC slightly increases during the rest of the Holocene while the reconstructions suggest a maximum in the mid-Holocene, followed by a weakening towards the Late Holocene (Supplementary Fig.S6).” Perhaps use the same terms as is shown on the figure. Here’s what I think the authors are saying: While the LOVECLIM and TraCE transient numerical simulations show a minimum around ~8ka followed by a steadily increasing AMOC through the rest of the Holocene, the reconstructed AMOC shows a similar low during the early Holocene but with a maximum in the mid-Holocene followed by a weakening towards the Late Holocene (Supplementary Fig.S6).

Absolutely. We fully agree with this phrasing.

In fact, it looks like the LOVECLIM is nearly opposite of the reconstructed AMOC at times (e.g. 6ka)...

Indeed, we agree.

Sometimes the LOVECLIM and TraCE are called transient numerical simulations and sometimes called full forcing experiments...perhaps use the same language throughout.

We used the term ‘full forcing’ in order to distinguish these simulations from those where an AMOC-correction had been applied. However, we changed some sentences and no consistently use the same terminology in the revised version.

S6 b and c: suggest using another color than blue for the AMOC corrected as it is confusing compared to S6a where the blue and black represented something different.

Yes, you are right. We appreciate this comment and have applied this correction accordingly.

Line 278: “in line with” is unclear. Perhaps “corresponding temporally with glacier advances. One thing is that while cooler temperatures and higher precipitation cause advance, the abandonment of the moraines is caused by retreat, so here where the cooler temperatures cause advance it is also important then that the temperature rises or precipitation decrease so that then the ice retreats and leaves a Late Holocene (or other) set of moraines.

We agree. Done.

Line 282: It isn't clear what "this" and "those" refer to exactly. I don't think this sentence is really necessary.

We replaced 'this' and 'those' with the correct terms. We believe, however, that this sentence is important, especially in the light of comments from R2.

Fig 1: It seems that "AMOC pattern" should be "TANAR pattern" – it is confusing if they are the same or different. Just pick one name for the pattern and stick to it (or are they different?) It seems the AMOC pattern here is described in the caption as: "a larger extent during the early Holocene than during the Late Holocene with substantial shrinking during the mid-Holocene." Which is similar to what they described in their rebuttal as the TANAR pattern (see comment above for lines 81-87. In their rebuttal they say: "As their evolution shows a good correspondence with AMOC changes, we replaced TANAR pattern in Fig. 1 with AMOC pattern.", but I think this is confusing – maybe just SAY that in the caption but refer everywhere to the pattern as the TANAR pattern.

Indeed, we used the two terms to clarify the fact that TANAR shows a good correspondence with the AMOC fingerprint, but apparently this did not have the right effect. We therefore changed it again and only use 'TANAR' in the new version.

Fig 2: Hopefully all of the figures will be larger – it is too small to read at this scale.

Yes – we are sure this can be addressed during copy-editing.

Fig 3: Suggest making numbers within circles bold for easier reading.

Done.

References used in this response:

- Hall et al., *QSR*, **28**, 2991–3009 (2009).
Kaplan et al., *Nature*, **467**, 194–197 (2010).
Kaplan et al., *Geology*, **41**, 887–890 (2013).
O'Hara et al., *Boreas*, **46**, 655–666 (2017).
Saha et al., *QSR*, **187**, 177–202 (2018).
Schweinsberg et al., *QSR*, **215**, 253–271 (2019).
Stroup et al., *Geology*, **42**, 347–350 (2015).
Vickers et al., *Geology* (2021) doi .org/10.1130/G48059.1
Young et al., *JQS*, **24**, 677–689 (2009).

Reviewer #2 (Remarks to the Author):

As most of the comments from Reviewer 2 (R2) do not require any actual change to our manuscript, we instead focused on their erroneous arguments which motivated our appeal.

Here we added complementary information to our appeal letter. To avoid any misunderstanding, in the revised manuscript we made the following changes.

1. The “*major concern is that the AMOC variations have minor impact on the Holocene temperature evolution of the Northern Hemisphere if compare with the external insolation change*”.

As explained in the appeal letter, we never considered insolation to be a minor factor. To further clarify this issue, we made the following changes:

We modified the title: **AMOC control on millennial-scale tropical glacier changes during the Holocene**

We changed a sentence in the abstract: **AMOC weakening favored a $\sim 0.5^{\circ}\text{C}$ temperature decrease north of 30°N , adding a substantial forcing on top of insolation changes,**

We also want to recall that in the previous version we wrote on lines 187-191:

“We propose that the AMOC also played a significant role on temperature during the mid- and late Holocene in the mid- and high-latitudes of the northern hemisphere superimposed on summer insolation that reached a minimum during the last millennium”.

Furthermore, the reviewer used one of the figure we provided to support his criticism. Nevertheless, while we do not claim that insolation is not playing a key role in the submitted manuscript, we believe that this figure was misinterpreted. On this figure, it is clear that the AMOC is not playing a dominant role for temperature north of 30°N , but it is still a key contributor to temperature variations. Indeed, as compared to TraCE simulation, the ratio of standard deviation between the signal due to AMOC and the one due to other external forcing reaches 45%, while it is of 58% when compared to LOVECLIM simulation. The variance explained of the total AMOC-adjusted by the AMOC component is also quite large (0.39 and 0.41 in TraCE and LOVECLIM respectively) and therefore not negligible, although lower as compared to the ones of external forcing.

Last but not least, the root mean squared error and the correlation with Kaufmann et al. (2020) reconstruction of summer temperature is improved by the correction of the simulations by the AMOC reconstruction. Correlations with reconstruction is improved from 0.64 to 0.73 in TraCE when the AMOC reconstruction is added, and from 0.51 to 0.74 in LOVECLIM, while RMSE is only marginally improved in both cases. Thus, even though the AMOC is not outperforming the role of external forcing in this region, as we said in the submitted manuscript, it is still a crucial contributor of variations of JJA temperature north of 30°N , and might have affected the glacier evolution in that respect. Furthermore, while the paper was correctly highlighting this fact, it puts also a large focus on the tropical area, which is a very wide region of the world, with lots of glaciers, and that also deserve attention.

2. R2 then writes: “*Summer insolation anomaly is on the order of 40 Watt/m^2 [...]strongest CO_2 emission projection (RCP8.5) can only produce 8.5 Watt/m^2 radiation anomaly*”.

In addition to what we wrote in our appeal, we added this sentence on line 299 which summarizes in a simple manner the number described in the latter point:“**In summer AMOC may represent about 1/3 of the cooling trend during the late Holocene in the Northern Hemisphere while 2/3 are primarily the result of insolation**”.

3. *Moreover, insolation change, in particularly, the precession change can largely affect the position off ITCZ, as Holocene precession anomaly could lead to warming the Northern Hemisphere but cooling the south.*

In addition to our rebuttal of R2's misconceptions explained in our appeal, we wish to emphasize what we actually wrote in our previous version of the manuscript. Already addressing above comment by the reviewer. See figure 2 curve f and lines 202-208: "This relationship between North Atlantic SST and tropical Andean glacier mass balance hinges on the migration of the Intertropical Convergence Zone (ITCZ), which is modulated by the AMOC strength and the associated inter-hemispheric SST dipole. Cold North Atlantic SST lead to an intensification of the South American summer monsoon (SASM), as the ITCZ, which serves as the major moisture conduit for the SASM, is displaced to the south, thereby enhancing moisture influx and convergence over tropical South America^{39,40}, thus leading to positive glacier mass balance." As well as lines 223-226 "AMOC also shifted the ITCZ position southward, enhancing related precipitation over the tropical Andes that favored large tropical glacier extents in the early and late Holocene and cooler temperatures due to positive land-surface feedbacks¹³, thus producing the overall TANAR signature".

Reviewers' Comments:

Reviewer #3:

Remarks to the Author:

I was asked to provide a review of this manuscript after it had been through more than 1 round of reviews already. To comment on whether the authors have presented sufficient evidence to show a robust mechanistic link between AMOC and variations in tropical glaciers.

To this specific question, my answer is no, I think they have presented an interesting and important idea or hypothesis for future studies to test. I am not an expert of AMOC and its impacts on the tropics. However, in my opinion, I think carrying out hosing experiments and then building that result into their reliance on model findings provides an interesting and worthy causal idea for future studies to think about or to test.

However, it seems to me though that the discussion between Rev2 and the authors is getting away from what I consider a worthy or noteworthy contribution of the manuscript in general, regardless of how the authors present AMOC's role as a cause. To step back and say that the paper merits consideration given, on its own 1) their original data and compilation of records. And they addressed many of reviewer's 1 concerns how this was done and 2) pointing out the interesting pattern they find (Fig. 1). It appears Reviewer 1 had concerns how they presented this, which they seemed to address (although not completely – see below).

That is, the authors show the tropics and North Atlantic sector have a strong linkage no matter what the ultimate cause. My suggestion is to separate out slightly better the 1) pattern they find (TANAR) which is a nice contribution and worthy itself and 2) AMOC's role as an idea or hypothesis they explore and it works for them. Right now, it does come across as this is the robust mechanistic link. I do not think this is necessary to make or break the worthiness of this publication. They can leave it more open, or at least exactly how AMOC would do this. Or, just emphasize more that whatever the cause, it must relate to a strong North Atlantic-tropical climate link, whether it is AMOC or some atmospheric mechanism or the tropics playing a smoking gun role.

I also have some issues with AMOC driving the tropics, although they seem to be different than those of Rev2, and I think it should NOT prevent the manuscript from publication. An important point is that I agree with the authors, in that local low latitude or mid latitude insolation cannot explain everything. The author's important finding (worthy of publication) is compelling that there seems to be a low latitude- North Atlantic pattern that is different from other parts of the globe (although there are still spatial gaps that need filling in). I have assumed something similar although I have not done the detailed work the authors have done nor had the original data from low latitudes as they include in this manuscript. Hence, I like their case for a strong tropical-North Atlantic link, no matter what the exact mechanism.

So, I think the manuscript works without AMOC being the main cause (and Rev2 issues). The manuscript does seem AMOC-centric. I read the manuscript quickly, and it does come across as they dismiss a stronger role for a tropical cause with a few sentences discussing just ENSO or other mechanisms. I suggest they also look at Chiang 2009 and Denton et al. 2010 (followed by 2021) or other papers, for example which also present more of an atmospheric sea-ice-atmospheric-ITCZ southward mechanism to link the two regions (Carr and Anderson 2010 had a nice write up in a Perspective summarizing). This problem will not be answered with this 1 paper.

My suggestions:

I assume the authors might feel a bit punch drunk after a few reviews, but if allowed to revise, I provide suggestions, some of which may overlap with some of Rev 2's issues. These can be addressed with rewriting.

1) given above, I think they are up against a bit of a chicken or egg problem, does AMOC change

happen first, or is AMOC part of a larger ocean-atmosphere climate response (see comment above) including what they state in Lines 316-317:

Line 316. Yet the drivers of the AMOC dynamics for this period are still poorly understood. AMOC
Line 317 changes may be related to external forcings, meltwater input⁴², or other missing processes,

Hence, my suggestion above to separate the findings a little bit better than they already do. That is, present it more as a testable idea or hypothesis, independent of the robustness of the pattern they see on Figure 1. Over the last ~10 years, some great records have come out that they include in this manuscript, including their original data, which allows them to discern such patterns on Figure 1.

2) I do not really follow the temperature vs precip discussion. I know this is might be secondary, but my understanding is summer temperature (or both) is dominant. They should provide a small paragraph with adequate references why they think glaciers are dominated by precipitation. I can equally cite a few studies that show summer conditions (or sunny-ness) or temperature are dominant. For example, the fact that the same finding is in the northern and southern Tropical latitudes tells me that it is temperature decrease that is key as that is more homogeneous than precipitation, which is more spatially variable (work of Oerlemans etc). When the sun is shining in summer, there is more ablation, conversely, when it is snowing radiative absorption will be less especially at these high elevations. At the very least temperature and precipitation change go hand-in-hand (e.g., go up to a glacier on a sunny day versus a snowy day and decide which feels colder).

However, I say secondary, because, again 1) their pattern on Figure 1 is robust no matter the definitive answer to this precip vs temp debate which predates their manuscript. And, 2) it seems to me that their idea that the North Atlantic and tropics are linked strongly (whether AMOC impacts the tropics?) shows up whether it is both temp and precip together or temperature dominate, which they do say in 1 or 2 places.

Other:

1) I think in figure 1 – which is sorts of results still, they should not call it the AMOC pattern (legend). This figure should provide a descriptive name. Perhaps just North Atlantic pattern. Or just use their TANAR as a label – that is more descriptive.

It seems Reviewer 1 had a similar comment. The authors responded "Indeed, we used the two terms to clarify the fact that TANAR shows a good correspondence with the AMOC fingerprint, but apparently this did not have the right effect. We therefore changed it again and only use 'TANAR' in the new version."

I may be missing something, but I do not see they changed this on Figure 1 – I see AMOC in the legend.

2) Even with a relatively quick glance, I noticed in the Table under country they wrote Chili. Chile is a food, not a country. Also, as far as I know, some of these studies are in Argentina, not Chile (nor Chili). The authors need to check the table.

Reviewer 3

That is, the authors show the tropics and North Atlantic sector have a strong linkage no matter what the ultimate cause. My suggestion is to separate out slightly better the 1) pattern they find (TANAR) which is a nice contribution and worthy itself and 2) AMOC's role as an idea or hypothesis they explore and it works for them. Right now, it does come across as this is the robust mechanistic link. I do not think this is necessary to make or break the worthiness of this publication. They can leave it more open, or at least exactly how AMOC would do this. Or, just emphasize more that whatever the cause, it must relate to a strong North Atlantic-tropical climate link, whether it is AMOC or some atmospheric mechanism or the tropics playing a smoking gun role.

- We fully agree with R3 about considering the AMOC influence on glaciers during the Holocene as a hypothesis that needs further investigations, rather than a foregone conclusion. To address this suggestion we changed the title, the abstract and now consistently use a conditional form when we mention the influence of the AMOC on glacier variations. The title now focuses on the TANAR pattern and not on the AMOC forcing as recommended by R3. We propose “**In-phase millennial-scale glacier changes in the Tropics and North Atlantic regions during the Holocene**” as our new title. In the main text for instance, we write on line 197 “**As millennial-scale AMOC changes influence both temperature and precipitation, we test how this driver could influence the TANAR glacier evolution during the Holocene**”.

So, I think the manuscript works without AMOC being the main cause (and Rev2 issues). The manuscript does seem AMOC-centric. I read the manuscript quickly, and it does come across as they dismiss a stronger role for a tropical cause with a few sentences discussing just ENSO or other mechanisms. I suggest they also look at Chiang 2009 and Denton et al. 2010 (followed by 2021) or other papers, for example which also present more of an atmospheric sea-ice-atmospheric-ITCZ southward mechanism to link the two regions (Carr and Anderson 2010 had a nice write up in a Perspective summarizing). This problem will not be answered with this 1 paper.

We thank R3 for these interesting suggestions and references. In our response we would like to distinguish between the question related to the possible mechanisms linking tropical glaciers and glaciers located in the Atlantic regions and questions related to the robustness of climate simulations when compared with proxy records including moraine records. We fully agree that the cause must be related to a strong North Atlantic-tropical climate link. As suggested by the papers of Chiang (2009), and Denton et al. (2010), an AMOC weakening associated with a larger sea-ice extent in the North Atlantic leads to a southern shift of the ITCZ. The changes in ITCZ location strongly impact tropical hydrology, and therefore glaciers's mass balance. Based on this reasoning, on previous studies and on the results of the AOGCM simulations, we suggest that changes in AMOC strength during the Holocene could have impacted the location of the ITCZ. We also agree with R3 that further studies are needed to conclude about the AMOC influence on tropical glaciers. However, our results also show that temperature and precipitation from full forcing experiments, including changes in insolation, volcanic activity, sea ice... do not permit to explain the observed TANAR pattern unless the AMOC intensity is corrected.

My suggestions: I assume the authors might feel a bit punch drunk after a few reviews, but if allowed to revise, I provide suggestions, some of which may overlap with some of Rev 2's issues. These can be addressed with rewriting. 1) given above, I think they are up against a bit of a chicken or egg problem, does AMOC change happen first, or is AMOC part of a larger ocean-atmosphere climate response (see comment above) including what they state in Lines 316-317:Line 316. Yet the drivers of the AMOC dynamics for this period are still poorly understood. AMOC Line 317 changes may be related to external forcings, meltwater input, or other missing processes,

We fully agree with R3 about “chicken or egg problem” and again fully agree with the fact that one paper won't solve this important issue. As mentioned earlier, we are now using a conditional form to distinguish results from our hypothesis. We also want to make it clear that the goals of our paper are i)

to highlight the consistent, in-phase glacier changes across the TANAR region during the Holocene, ii) to link these changes to potential external drivers, iii) to suggest that the observed changes may have occurred in response to AMOC variations during the Holocene. Explaining the processes that led to such AMOC changes during the Holocene is, however, beyond the scope of this manuscript.

This is noted on L. 318:

“highlights the need for a better understanding of the causes of AMOC variations over the Holocene”. Yet the drivers of the AMOC dynamics for this period are still poorly understood. AMOC changes may be related to external forcings, meltwater input⁴³, or other missing processes, such as changes in Mediterranean outflow during the so-called Sapropel S1 event⁴⁸. »

Hence, my suggestion above to separate the findings a little bit better than they already do. That is, present it more as a testable idea or hypothesis, independent of the robustness of the pattern they see on Figure 1. Over the last ~10 years, some great records have come out that they include in this manuscript, including their original data, which allows them to discern such patterns on Figure 1.

As mentioned earlier we agree with this suggestion and changed our text accordingly.

2) I do not really follow the temperature vs precip discussion. I know this is might be secondary, but my understanding is summer temperature (or both) is dominant. They should provide a small paragraph with adequate references why they think glaciers are dominated by precipitation. I can equally cite a few studies that show summer conditions (or sunny-ness) or temperature are dominant. For example, the fact that the same finding is in the northern and southern Tropical latitudes tells me that it is temperature decrease that is key as that is more homogeneous than precipitation, which is more spatially variable (work of Oerlemans etc). When the sun is shining in summer, there is more ablation, conversely, when it is snowing radiative absorption will be less especially at these high elevations. At the very least temperature and precipitation change go hand-in-hand (e.g., go up to a glacier on a sunny day versus a snowy day and decide which feels colder). However, I say secondary, because, again 1) their pattern on Figure 1 is robust no matter the definitive answer to this precip vs temp debate which predates their manuscript. And, 2) it seems to me that their idea that the North Atlantic and tropics are linked strongly (whether AMOC impacts the tropics?) shows up whether it is both temp and precip together or temperature dominate, which they do say in 1 or 2 places.

At the global scale temperature and precipitation changes may both significantly impact glaciers. In extratropical regions, temperatures are generally well correlated with some variables of the energy budget (e.g., Sicart et al., 2008), and it is generally assumed that temperature changes largely controlled the long term glacier behavior during the Holocene. However, in the outer tropics, where Zongo glacier is located (i.e. the glacier used in this paper as a target to investigate relationships between climate conditions and mass balance; see Fig. 3 for instance), it is now well established that current glacier mass balance is mainly controlled by precipitation changes (Wagnon et al., 2001; Favier et al., 2004; Sicart et al., 2005). For instance Francou et al. (2003) showed that about 70% of mass balance variability is controlled by precipitation between September and March, whereas ablation and accumulation are limited during the rest of the year. In addition, several papers conducted on direct mass and energy balance measurements show that there is almost no relationship between temperature and melting at a seasonal and annual time scale in the outer tropics, suggesting for instance that the degree models are not adapted to analyze glacier changes in this region (Sicart et al., 2008). This is mainly due to the fact that there are only limited variations of temperature in the tropics at seasonal or annual time scale. Some of the papers cited here are already cited in the text (see for instance, Favier et al., 2004; Rabatel et al., 2013). However at an orbital time scale we assumed that temperature may have a minor influence on tropical glacier changes as identified in Jomelli et al. (2014). Since glaciers in the extratropical and in the tropical region respond differently to precipitation and temperature changes, we considered both variables in our analysis. We are showing that the

AMOC had an influence on both variables. The very strong effect on precipitation changes in the tropics was crucial to justify glacier changes in this region.

Other:

1) I think in figure 1 – which is sorts of results still, they should not call it the AMOC pattern (legend). This figure should provide a descriptive name. Perhaps just North Atlantic pattern. Or just use their TANAR as a label – that is more descriptive. It seems Reviewer 1 had a similar comment. The authors responded "Indeed, we used the two terms to clarify the fact that TANAR shows a good correspondence with the AMOC fingerprint, but apparently this did not have the right effect. We therefore changed it again and only use 'TANAR' in the new version." I may be missing something, but I do not see they changed this on Figure 1 – I see AMOC in the legend.

We agree with this suggestion and labeled the pattern as 'TANAR' in Fig. 1. However, there may be a slight misunderstanding related to this aspect. The term 'TANAR pattern' was used in the very first version of the draft. However, both reviewers R1 and R2 asked to change this term. R1 suggested using 'AMOC pattern' and R2 had yet another opinion. In order to find a solution that would satisfy both reviewers, we used the term 'AMOC' in the revised version and mentioned at the end of the sentence the following: "AMOC pattern (blue circles) refers to a larger extent during the early Holocene than during the Late Holocene with substantial shrinking during the mid-Holocene". As we also included some clarifications about the TANAR pattern in the supplementary text as requested by R1 in the first round of reviews, we hope that the term 'TANAR' is now clear enough to be used in Fig. 1.

2) Even with a relatively quick glance, I noticed in the Table under country they wrote Chili. Chile is a food, not a country. Also, as far as I know, some of these studies are in Argentina, not Chile (nor Chili). The authors need to check the table.

Oops! Thank you for this catch. Indeed, 'Chile' was the right word. We revised this table and corrected the location of some glaciers located in Argentina. Thank you very much for this remark.

References mentioned above

Favier, V., et al. Glaciers of the outer and inner tropics: a different behavior but a common response to climatic forcing, *Geophys. Res. Lett.*, 31, L16403, (2004).

Francou, B., et al. Tropical climate change recorded by a glacier in the central Andes during the last decades of the twentieth century: Chacaltaya, Bolivia, 16°S, *J. Geophys. Res.*, 108, 4154, (2003).

Jomelli, V. et al. A major glacial advance during the Antarctic Cold Reversal. *Nature* **513**, 224–228 (2014).

Rabatel, A. et al. Current state of glaciers in the tropical Andes: a multi-century perspective on glacier evolution and climate change. *The Cryosphere* **7**, 81-102, (2013).

Sicart, et al. Atmospheric controls of heat balance of Zongo Glacier (16°S, Bolivia), *J. Geophys. Res.* 110, D12106, (2005).

Sicart, et al. Glacier melt, air temperature and energy balance in different climates: The Bolivian Tropics, the French Alps, and northern Sweden, *J. Geophys. Res.*, 113, D24113, (2008).

Wagnon, et al. Annual cycle of energy balance of Zongo Glacier, Cordillera Real, Bolivia, *J. Geophys. Res.*, 104, 3907–3923, (1999).

Wagnon et al. Anomalous heat and mass balance budget of Glaciar Zongo, Bolivia, during the 1997/98, El Nino year, *J. Glaciol.*, 47, 21–28; (2001).

Reviewers' Comments:

Reviewer #3:

Remarks to the Author:

First, in one of the tables, I still see Chili, not Chile. In the 'Scaled length' worksheet. I would be remiss if I did not point this out.

More important, the authors have addressed my comments adequately.

However, I add two points the authors may want to address even more adequately or think about in their final edits.

1) The authors may want to look at and think about Denton et al. 2021, which has quite a different conclusion or thesis than the 2010 paper that the authors discuss including in their rebuttal. Although it is not clear to me how much this changes all of their main conclusions, given once the system is kicked it seems to be a chicken or egg issue as discussed in my review and they recognize in their rebuttal. Their conclusions and findings may be most pertinent after the smoking gun from the South and Southern Westerlies gets the climate system moving, as discussed in Denton et al 2021?

For example, perhaps edit lines 324-326 to include in addition (to what they have already) the Denton et al. 2021 idea, that the southern hemisphere westerlies are key and could also be a driver of the North Atlantic region and AMOC dynamics.

2) In their rebuttal, the authors addressed the issue of temperature and precipitation, that is to me and the Editor perhaps. I meant perhaps to strengthen or add to the manuscript some of the text and references they have in the rebuttal. Perhaps have an edited version of their "rebuttal paragraph" in or near the top of the Methods sections (i.e., Lines 369+). I may be missing it, with albeit a quick glance, some of these references are not cited, for example, Wagon et al. 1999 and 2001.

That is, do the authors want to add some of the text they put in the rebuttal including more references? Looking at other parts of the text, they do acknowledge temperature and precip change together in places. However, it seems to me some readers may want to see more references given this debate of temp vs precip (or both). I assume there is not a reference limit for the Methods Section?

For example, maybe Lines 202-203

"Recent AMOC reconstructions¹⁴⁻¹⁷ also suggest that the AMOC may have influenced IN PARTICULAR precipitation in the tropical Andes throughout the Holocene, thereby affecting glacier evolution (THEIR REFERENCES or this gets backed up in the Methods section?)

First, in one of the tables, I still see Chili, not Chile. In the 'Scaled length' worksheet. I would be remiss if I did not point this out.

- We are really sorry. Please forgive us for this mistake once again! Thank you for this double check

More important, the authors have addressed my comments adequately. However, I add two points the authors may want to address even more adequately or think about in their final edits.

1) The authors may want to look at and think about Denton et al. 2021, which has quite a different conclusion or thesis than the 2010 paper that the authors discuss including in their rebuttal. Although it is not clear to me how much this changes all of their main conclusions, given once the system is kicked it seems to be a chicken or egg issue as discussed in my review and they recognize in their rebuttal. Their conclusions and findings may be most pertinent after the smoking gun from the South and Southern Westerlies gets the climate system moving, as discussed in Denton et al 2021?

For example, perhaps edit lines 324-326 to include in addition (to what they have already) the Denton et al. 2021 idea, that the southern hemisphere westerlies are key and could also be a driver of the North Atlantic region and AMOC dynamics.

- We fully agree with the suggestion proposed by reviewer 3 see lines 323-328. Consequently we added the following sentences and reference: **Following the "Zealandia switch" hypothesis (Denton et al., 2021), a poleward shift of the SH westerlies between the early and mid-Holocene would have led to a global warming. We note that this poleward shift of the SH westerlies could have also led to an AMOC strengthening, partly through the modulation of the Aghulas leakage, in agreement with our suggested AMOC evolution through the Holocene.**

2) In their rebuttal, the authors addressed the issue of temperature and precipitation, that is to me and the Editor perhaps. I meant perhaps to strengthen or add to the manuscript some of the text and references they have in the rebuttal. Perhaps have an edited version of their "rebuttal paragraph" in or near the top of the Methods sections (i.e., Lines 369+). I may be missing it, with albeit a quick glance, some of these references are not cited, for example, Wagon et al. 1999 and 2001.

That is, do the authors want to add some of the text they put in the rebuttal including more references? Looking at other parts of the text, they do acknowledge temperature and precip change together in places. However, it seems to me some readers may want to see more references given this debate of temp vs precip (or both). I assume there is not a reference limit for the Methods Section?

- There were 2 papers already cited in the draft Favier et al., (2004) and Rabatel et al., (2013) but there were not mentioned while we discussed the question of precipitation. Unfortunately we already cited the maximum possible number of references which is 70. So we removed one reference to include Denton et al (2021) but we cannot include the other references mentioned in our rebuttal. So we just mentioned Favier et al., 2004, and Rabatel et al. 2013 (which were already cited). But again thank you very much for this proposition. We created a new paragraph in the method section see line 557 and included the following text:

Links between temperature/precipitation and tropical mass balance.

At the global scale temperature and precipitation changes may both significantly impact glaciers. In extratropical regions, temperatures are generally well correlated with some variables of the energy budget (e.g., Rabatel et al., 2013), and it is generally assumed that temperature changes largely controlled the long term glacier behavior during the Holocene. However, in the outer tropics, where Zongo glacier is located (i.e. the glacier used in this paper as a target to investigate relationships between climate conditions and

mass balance; see Fig. 3 for instance), it is now well established that current glacier mass balance is mainly controlled by precipitation changes (Favier et al., 2004; Rabatel et al., 2013). For instance recent observations showed that about 70% of mass balance variability is controlled by precipitation between September and March, whereas ablation and accumulation are limited during the rest of the year. In addition, several papers conducted on direct mass and energy balance measurements show that there is almost no relationship between temperature and melting at a seasonal and annual time scale in the outer tropics. This is mainly due to the fact that there are only limited variations of temperature in the tropics at seasonal or annual time scale. However, on longer time scales (e.g., multi-decadal, centennial, millennial), increase in air temperature may also have an influence on tropical glacier changes as identified in Jomelli et al. (2014). Temperature may impact precipitation phase on tropical glaciers, through changes in the elevation of the 0°C isotherm and therefore of the rain/snow limit with impacts on both the snow accumulation amount and the glacier surface albedo (Favier et al., 2004). Thus, in spite of a major control of precipitation on tropical glacier changes, long term significant warming or cooling are also critically transferred into glacier mass loss (or gains).

For instance line 223 "Recent AMOC reconstructions 14-17 also suggest that the AMOC may have influenced IN PARTICULAR precipitation in the tropical Andes throughout the Holocene, thereby affecting glacier evolution (THEIR REFERENCES or this gets backed up in the Methods section?)

- As mentioned earlier we cannot add references here so we mentioned the method section.